



# Extended validation and evaluation of the OLCI-SLSTR Synergy aerosol product (SY_2_AOD) on Sentinel-3

Larisa Sogacheva[1], Matthieu Denisselle[2], Pekka Kolmonen[1], Timo H. Virtanen[1], Peter North[3], Claire
Henocq[2], Silvia Scifoni[4] and Steffen Dransfeld[5]

[1] Finish Meteorological Institute, Climate Programme, Helsinki, 00540, Finland
[2] ACRI-ST, Sophia-Antipolis, 06410, France
[3] Global Environmental Modelling and Earth Observation (GEMEO), Dept. of Geography, Swansea University, SA28PP, UK
[4] Serco Italia SpA for European Space Agency (ESA), European Space Research Institute (ESRIN), 00044 Frascati, Italy.
[5] European Space Agency (ESA), European Space Research Institute (ESRIN), Frascati, Italy

*Correspondence to*: Larisa Sogacheva (larisa.sogacheva@fmi.fi)

**Abstract**

We present the first extended validation of a new synergy global aerosol product (SY_2_AOD) which is based on synergistic
use of data from the Ocean and Land Color Instrument (OLCI), and the Sea and Land Surface Temperature Radiometer
(SLSTR) sensors onboard the Copernicus Sentinel-3A (S3A) and Sentinel-3B (S3B) satellites. Several approaches, including
statistical analysis, time series analysis, comparison with similar aerosol products from the other spaceborne sensor Moderate
Resolution Imaging Spectroradiometer (MODIS), were used for validation end evaluation of S3A and S3B SY_2 aerosol
products, including Aerosol Optical Depth (AOD) provided at different wavelengths, AOD pixel level uncertainties, fine mode
AOD and Angström exponent.

Over ocean, the performance of SY_2 retrieved AOD is good: for S3A and S3B respectively, correlation coefficients with the
Maritime Aerosol Network (MAN) component of the AErosol RObotic NETwork (AERONET) are 0.88 and 0.85; 88.6% and
89.5% of pixels fit into MODIS error envelope of ±0.05±0.2*AOD.

Over land correlation coefficients with AERONET AOD are 0.60 and 0.63 for S3A and S3B respectively; 51.4% and 57.9%
of pixels fit into MODIS error envelope. Reduced performance over land is expected since the surface reflectance and angular
distribution of scattering is higher and more difficult to predict over land than over ocean. The results are affected by a large
number of outliers.

Evaluation of the per-retrieval uncertainty indicated good correlation with measured error distributions. The retrieval of
Angstrom exponent, related to aerosol size distribution, shows good spatial correlation with expected sources but generally
overestimates AE for cases where AERONET Angstrom is low, resulting in overall high bias.



Differences between the annual and seasonal AOD values from SY_2 and MODIS Dark Target and Deep blue products are within 0.02 for the study area. The agreement is better over ocean. Over bright land surface (Saharan desert) the difference between two products is highest (up to 0.11), and the sign of the difference varies over time and space.

For both S3A and S3B AOD products, validation statistics are often slightly better in the Southern hemisphere. In general, the
performance of S3B is slightly better.

## 1 Introduction

The concern about climate change (e.g., Bergquist and Warshaw, 2019) along with a willingness to reduce its effects (e.g., Leiserowitz et al., 2020; Hoffmann et al., 2022) are of growing interest during the past decades. Global models introduce different scenarios for climate change (Arbor et al., 2021; Meehl et al., 2007), which are often based on the historical records
and trends. Satellite data provide unique global data on the Earth's surface and atmosphere (including aerosols); they are assimilated into global and regional models (Khaki et al., 2020; Eyre et al., 2022) and used for model evaluation (Gliß et al., 2021).

With the fast development of the space-born instruments, including improved quality of onboard instruments and increased temporal and spatial coverage (CEOS, 2017; Dubovik et al., 2021), and on the other hand with improved access to satellite
products (Borowitz, 2018) following open access policy (Harris and Bauman, 2015; Olbrich, 2018) and standardisation of satellite data (Loew et al., 2017), the contribution of the space-borne measurements in climate studies is gradually increasing. Despite having an advantage in coverage over ground-based products, satellite products often concede lower quality. Product quality is an important part of satellite mission requirements.

Calibration and validation (cal/val) are essential to characterise the quality of the performance of a mission
(https://earth.esa.int/eogateway/documents/20142/1564943/Sentinel-3-Calibration-and-Validation-Plan.pdf, last access 14.02.2022). Calibration tasks include pre-launch and in-flight calibrations and characterisation, as well as comprehensive verification of Level-1 data processors. For optical missions, radiometric, spectral, and geometric stability are subjects for investigation.

Validation is a part of a cal/val activity. In the context of remote sensing, validation refers to the process of quantifying the
accuracy of satellite retrieved products by assessing the uncertainty of the derived products by analytical comparison to reference data, which is presumed to represent the true value of an attribute. Validation shows the maturity of the satellite derived product and, thus, provides a conclusion on the mission success. Besides providing information about the product quality, validation may reveal a degradation of the instrument or potential drift (Julien and Sobrino, 2021). Validation results should be used in quality assurance reporting together with product details, calibration characterisation, retrieval algorithm
description, and uncertainty characterisation.



Validation is a comparison against in-situ measurements, systematic and campaigns, and inter-comparison against other satellite data sources and/or models. However, since other satellite products and models may have their own bias, the inter-comparison against other satellite products is called evaluation.

General validation is product-specific, while detailed validation is instrument-specific. Common validation objectives, approaches and challenges should be applied to allow inter-comparison of validation results and thus, inter-comparison of similar products. Changes in sensors and algorithms may be revealed if the same validation approach is applied. However, validation approaches have to be adapted considering specifications of particular products (e.g., temporal, spatial, radiometric resolutions).

Validation requires reference data with high reliability. Possible uncertainties of the product used as the "truth" must be considered. Since the performance of a retrieval algorithm may vary in different conditions, validation also requires well-sampled coverage of useful ranges of measured values.

An independent verification processing system is important. Validation requires an expertise on instrument, processing, and application, and a good understanding of limitations. Iterations on the product validation results with product developers, such as the round robin approach (Holzer-Popp et al, 2013), should be considered to better understand the results. The purpose of validation is not only to show how good or bad the product is; issues explaining differences between product and reference data should be identified. Based on validation and evaluation results, recommendations on the product improvements can be provided to the product developers. Recommendations are important as they will help to identify conditions where an algorithm performance should be improved.

In this paper we introduce global validation and evaluation results for the Synergy Aerosol Optical Depth (AOD) product SY_2_AOD (North and Heckel, 2019) for the period 01.2020-09.2021. The SY_2_AOD product is retrieved from spatially and temporally collocated data measured with two instruments, Sea and Land Surface Temperature Radiometer (SLSTR) and Ocean and Land Color Instrument (OLCI) onboard Sentinel-3 (S3A and S3B) satellites. The synergy retrieval algorithm has been originally developed for the retrieval of AOD from the Advanced Along-Track scanning Radiometer (AATSR) and MEdium-spectral Resolution Imaging Spectrometer (MERIS) (North et al., 2008) and further developed for the S3 instruments. The SY_2_AOD product is available from 15.01.2020 from both S3A and S3B satellites. Extensive and systematic AOD validation against ground-based measurements and inter-comparison with Moderate Resolution Imaging Spectroradiometer (MODIS) AOD product were performed in the frame of the European Space Agency (ESA) "ESA/Copernicus Space Component Validation for Land Surface Temperature, Aerosol Optical Depth and Water Vapour Sentinel-3 Products" (LAW, https://law.acri-st.fr/home, last access 10.01.2022).

The paper is structured as following. The SY_2 retrieval algorithm and SY_2_AOD product are introduced in Sect.2. In Sect. 3 we introduce a validation approach applied in the current study. An algorithm developed for extracting satellite and ground-based measurements matchups is explained in Sect.4. Reference validation products are introduced in Sect.4. AOD, AOD uncertainties, Fine mode AOD (FMAOD), Fine Mode Fraction (FMF), Angström exponent (AE) validation results with AERONET are shown in Sect. 6-8. Validation results over ocean are presented in Sect. 9. $AOD_{550}$ validation results with



SURFRAD and SKYNET are shown in the Supplement (Sections S1and S2, respectively). Inter-comparison of daily, monthly, seasonal, and annual SY-2 AOD and MODIS AOD products is shown in Sect.10. Validation results are summarised in Sect. 11.

## 2    SY_2 AOD product

### 2.1    Instrument description

OLCI and SLSTR L1b top-of-the-atmosphere (TOA) radiances were utilized in the SYNERGY algorithm for the retrieval of aerosol properties.

The SENTINEL-3 OLCI (https://sentinels.copernicus.eu/web/sentinel/technical-guides/sentinel-3-olci/olci-instrument, last access 16.03.2022) is a push-broom imaging spectrometer with a swath width of 1270 km. It provides spatial sampling at 300 m with five cameras in 21 bands in the spectrum range of 0.4-1.2 μm.

The SLSTR instrument (https://sentinels.copernicus.eu/web/sentinel/technical-guides/sentinel-3-slstr/instrument, last access 16.03.2022 ) is a conical scanning imaging radiometer employing the along track scanning dual view technique. With the dual view scan (at near nadir and 55° oblique), measurements are taken at nine bands in the range of 0.55-12 μm covering the visible, shortwave infrared, and thermal infrared areas of the spectrum. The SLSTR spatial resolution is 500m at nadir for visible and shortwave infrared bands; 1km at thermal infrared.

### 2.2    Algorithm description


The aim of the SYNERGY aerosol algorithm is to provide global aerosol optical depth and related aerosol properties for all cloud and ice-free regions of the Sentinel-3 combined OLCI / SLSTR instrument swaths. The SLSTR retrieval (ESA Aerosol CCI portal, https://climate.esa.int/en/projects/aerosol/key-documents/, Product Validation and Intercomparison Report, last access: 25.02.2022) is of variable quality, with higher uncertainty in retreievals in the oblique backscattering direction. The

motivation of combining with OLCI is to improve on the SLSTR retrieval especially in these regions using additional spectral information from OLCI. The algorithm is derived originally from the aerosol retrieval algorithm developed by Swansea University under the ESA Aerosol CCI programme for the (A)ATSR and SLSTR instruments (North 2002; Bevan et al., 2012; Popp et al., 2016) but with further development to exploit the increased spectral sampling available from the OLCI instrument. This aims to allow a more robust retrieval, but also to provide aerosol estimates over the full Sentinel-3 swath, whereas for the

original algorithms using only SLSTR imagery, retrieval over land is only attempted for the regions where both nadir and oblique views are available. The key features of the algorithm are given here and are summarised in detail the SYN AOD Algorithm Theoretical Basis Document (North and Heckel, 2019).



### 2.2.1 Pre-processing

The algorithm uses the L1c co-registered OLCI and SLSTR data product as input, projected on the OLCI grid. Co-registration
is made based on the common 865 nm radiometric band. Over selected ground-control points, radiometric images
of SLSTR 865 nm band are extracted and compared to the OLCI 865 nm acquisitions. The OLCI image is moved
around according to shift vectors and the cross-correlation with the fixed SLSTR window is calculated. The
elements of the shift vectors at which a maximum in cross-correlation is reached determine the pixel deregistration
between OLCI and SLSTR reference channel.

Over ocean, AOD is returned using the full swath of the L1c product (1400 km), while over land the region covered by both
nadir and oblique view (750km) is used for best quality retrieval, and aerosol retrieval is also made outside of this region where
both nadir-only SLSTR and OLCI is available (~1200 km). Beginning with the L1c product, pixels are flagged to screen cloud,
snow ice or sun glint areas. In addition, all neighbouring pixels to cloud pixels are flagged to avoid edge effects. Pixels are
grouped into 'super-pixels' formed by blocks of 15x15 pixels of the L1c SYN pixels at 300m spatial resolution. Thus a super-
pixel represents a resolution of about 4.5 km x 4.5 km. The result is a super-pixel giving aggregated cloud-free TOA radiance
for nadir and oblique view (if present) of the same surface location. Over ocean retrieval proceeds if either nadir or oblique
super-pixels are valid (i.e., formed from at least 50% of valid pixels), while over land both nadir and oblique must be valid for
dual view retrieval, or nadir only for single view (spectral) retrieval.

### 2.2.2 Inversion to derive aerosol parameters

The basis of the algorithm is iterative non-linear optimisation to jointly retrieve aerosol optical depth at a reference wavelength
of 550nm, referred to as $AOD_{550}$, and fine mode fraction of $AOD_{550}$ (FM AOD). Atmospheric radiative transfer is approximated
as a Look-up Table (LUT) to relate top of atmosphere to surface reflectance, for a given estimate of aerosol parameters, water
vapor, ozone and surface pressure. Over both land and ocean, the retrieval requires optimisation of a cost function expressing
fit of derived surface reflectance to ocean or land models of reflectance. The inversion is carried out for all land and ocean
super-pixels which are at least 50% free of cloud, ice and snow. Several additional parameters are provided, derived from these
properties, to provide information on spectral variation of AOD, and surface reflectance values intended as diagnostics (Table
S1, Supplement). Where a single direction is used, the inversion is made over spectral bands in that direction only. This is
normally the case outside the oblique view swath, where nadir only is used, but use of the oblique view alone also occurs over
ocean where the nadir view is obscured by glint or cloud. Over ocean, only SLSTR channels (five spectral bands, corresponding
to S1 (554 nm); S2 (659 nm); S3 (865 nm); S5 (1613 nm) and S6 (2255 nm)) are taken into account in the aerosol retrieval.
Over land, both sensors (including OLCI 442.5 spectral band) are considered.

A climatology of aerosol composition (Kinne et al., 2013; de Leeuw et al., 2015) is used to provide further aerosol composition
information of the fine and coarse components (non-spherical vs spherical, single scattering albedo) and a prior estimate of





fine mode fraction. Further constraints prevent unfeasible retrieval (e.g. negative AOD or surface reflectance). An estimate of the 1 standard deviation (std) error in AOD at 550 nm is derived from the second derivative (curvature) of the error surface near the optimal value.

Over ocean, a surface reflectance model gives a reflectance estimate determined from the wind speed and direction and using the models of Cox and Munk (1954) for glint, Monahan and O'Muircheartaigh (1980) and Koepke (1984) for foam fraction

and spectral reflectance, and Morel's case I water reflectance model dependent on pigment concentration (Morel, 1988). The ocean inversion uses bands from SLSTR only, using both views to invert if both are available, or a single view (either nadir or oblique) where one view is either obscured by cloud, is contaminated by glint, or lies in a swath region where only a single view is present. For land, the reflectance constraint is the result of fitting to separate angular and spectral parameterised models (North, 2002; North et al., 2008; Davies and North, 2015; North and Heckel, 2019). Where the oblique SLSTR view is not

available, only the spectral constraint is used, allowing AOD estimation over the full L1c swath over both land and ocean.

### 2.2.3    Post-processing

A final step is used to filter residual cloud contamination or other sources of poor retrieval. This is based on thresholding of local image standard deviation, discussed in Sogacheva et al., 2017. Over ocean a final screening is also made on the quality of model fit. Any AOD value outside the AOD valid range of [0, 4] is replaced by a 'fill' value 6.53. 'Clean-air' test is

performed to recognise cases when an extensive rejection of low AOD values occurs in case of clean atmosphere, which often happens over dark surfaces. In case this test is positive, which is indicated in quality flags, a value of 0.04 is used.

During post-processing, derivation of further aerosol outputs from the retrieved $AOD_{550}$ and FM AOD, such as Angstrom exponent, is performed. Full set of quality flags is provided.

### 2.3    SY_2 AOD product description

Derived aerosol outputs include AOD, AOD uncertainty and single scattering albedo (each at 440nm, 550nm, 670 nm, 865 nm, 1610 nm), aerosol absorption optical depth, fine mode AOD, duct AOD (each at 550nm) and Angström exponent (between 550nm and 865nm). The full list of derived aerosol outputs which are recorded in gridded NetCDF format at 4.5km resolution, is shown in Table S1. Additionally for each super-pixel, information is provided giving time and location, solar/view geometry, cloud fraction, AOD retrieval quality flags, and retrieved surface reflectance for each waveband. Quality flags indicate which

retrieval method was used, for example nadir-only or dual view, land/ocean algorithm and further indicators such as retrieval failure through negative AOD estimation or glint contamination.

### 3    Validation approach

The validation approach suggested for the European Space Agency (ESA) Climate Change Initiative (CCI) AOD product validation (ESA Aerosol CCI portal, https://climate.esa.int/en/projects/aerosol/key-documents/, Product Validation and



Intercomparison Report, last access: 25.02.2022; de Leeuw et al., 2015) and currently used in ESA Aerosol CCI and Copernicus C3S_Lot2 projects was followed. A similar validation approach has been applied and further developed in Sogacheva et al. (2018a, 2018b, 2020) for validation of the AATSR, MODIS and merged AOD products. The approach includes three main steps: i) match-up between satellite-retrieved AOD and ground-based measurements (Sect.4), ii) statistical tools application to the set of matchups to reveal the agreement between two products (Sect.6) and iii) analysis of the statistics.

Different aspects of the validation and evaluation of various AOD products (Chu et al., 2002; Ichoku et al., 2002; Remer et al., 2005; Levy et al., 2013; Shi et al., 2013; Sayer et al., 2012a,b, 2013, 2018, 2019) have been considered. Analysis of the AOD pixel-level provided uncertainties was performed based on the recommendations by Sayer et al. (2020) and considering best practices from ESA Aerosol CCI.

Annual and seasonal validation was performed globally for all data, and additionally over land and ocean, respectively.

Furthermore, respective validations were made over selected areas, which represent different surface and aerosol types.

Since the back scatter contribution to the radiance measured at the top of the atmosphere is more critical in the NH (e.g., https://www-cdn.eumetsat.int/files/2021-09/SARP_Report_Option_1_final.pdf) and difference in AOD quality between the Northern Hemisphere (NH) and Southern Hemisphere (SH) for the SLSTR products has been revealed earlier (https://climate.esa.int/media/documents/Aerosol_cci_PVIR_v1.2_final.pdf), SY_2 AOD products from the NH and SH were

validated separately.

$syAOD_{550}$ validation was performed for all available matchups and separately for groups of the matchups assorted based on prevailing aerosol types. Aerosol types were defined with AERONET AOD (aAOD) and AERONET AE (aAE) thresholds. Although these thresholds are subjective, we consider "background" aerosol to be cases where $aAOD_{550} < =0.2$, "fine-dominated" with $aAOD_{550} > 0.2$ and $aAE > =1$, and "coarse-dominated" with $aAOD_{550} > 0.2$ and $aAE < 1$ (e.g. Eck et al.,

1999). This classification has also been used by e.g. Sayer et al. (2018) and Sogacheva et al. (2018a, b, 2020).

Another specification of the SY_2 AOD product is that the AOD retrieval has been performed with different retrieval approaches, depending on SLSTR and OLCI coverage and L1B data availability in different viewing angles (for details, see Sect.2). Dual-view processor has been applied when SLSTR measurements from both views, nadir and oblique, were available. If measurements were available from one view only, the single view processor was applied to either nadir (over either land or

ocean) or oblique view (over ocean or inland waters only). This specification of the product was considered in the current validation exercise.

## 4    Matchups

A matchup is defined as the combination of simultaneous and spatially collocated satellite and ground-based measurements. Following Ichoku et al (2002), a macro pixel of 11 *11 SY_2 AOD pixels (a surface of ca 50km*50km) around each station

was extracted at each overpass over a ground-based measurement station. All ground-based measurements were acquired in a





time window of +/- 30 minutes around the satellite crossing time were considered. Statistics such as number of measurements, mean, median, minimum, maximum and standard deviation computed over this time frame were included in the matchup files. All ground-based measurements were extracted from well-qualified networks introduced in Sect. 5.1 (AERONET) , Sect. 5.2 (MAN) and in the supplement (SURFRAD, SKYNET); no additional quality control check has been performed for the

reference data. On the contrary, all satellite extractions included all quality flags and contextual parameters presents in the Sentinel 3 operational products. Satellite extractions were created automatically for each station, at each overpass, and centred on the station location. They were then associated with relevant ground-based measurements when these data were available and validated.

"Empty" matchups, i.e., when the whole satellite extraction is associated with a fill value for AOD, were not filtered out from

the database, except in case of operational issues in the Sentinel-3 instruments. As these fill values were mainly due to cloud contamination or aerosol retrieval failure, they may provide information about the performance of, e.g., cloud screening in the SY_2 algorithm and were therefore relevant to validation objective.

A free access (upon subscription) to this matchups database has been provided on the ESA LAW web portal (https://law.acri-st.fr/home, last access 10.01.2022).

To explore the performance of different processors, four separate datasets were created and validated separately. The first dataset (called 'all' in the following) consists of all available data, regardless of which processor was used. The second dataset ('dual') contains data retrieved with the dual view processor. The third ('singleN') and fourth ('singleO') dataset are created using the single view processors applied to nadir or oblique views, respectively. The total number of matchups from dual, singleN and singleO groups is higher than the total number of 'all' matchups, because in 11*11 pixels area around reference

ground-based measurement there could have been pixels retrieved with different processors (e.g., dual and singleN). In that case we have two matchups (one for dual group and one for single group) for the same spatio-temporal window. If the group not mentioned specifically ('dual', 'singleN' or 'singleO', in the text or in the figure), results are shown and discussed for the group 'all'.

## 5    Reference datasets

### 5.1    AERONET

The AERONET is a federation of ground-based remote sensing aerosol networks (https://aeronet.gsfc.nasa.gov/). For more than 25 years, AERONET has provided a long-term, continuous, and readily accessible public domain database of aerosol optical, microphysical, and radiative properties for aerosol research and characterization, validation of satellite retrievals, and synergism with other databases. An extensive description of the AERONET sites, procedures and data provided is available

from the AERONET web site and in (Holben et al., 1988, Giles et al., 2019).



Ground-based sun photometers directly observe the attenuation of solar radiation without interference from land surface reflections. They provide accurate measurements of AOD with uncertainty ~0.01–0.02 (Eck et al., 1999) in the spectral range of 340-1640 nm.

For the AOD validation, AERONET version 3 data (Giles et al., 2019) – automated near-real-time quality control algorithm
with improved cloud screening for Sun photometer aerosol optical depth (AOD) measurements - was be utilized. Version 3 AOD data are computed for three data quality levels: Level 1.0 (L1.0, unscreened), Level 1.5 (L1.5, cloud-screened and quality controlled), and Level 2.0 (L2.0, quality-assured). The Level 2.0 AOD quality-assured dataset is now available within a month after post-field calibration, reducing the lag time from up to several months.

Since AERONET is a network of ground-based sun-photometers, and while some of the AERONET stations are in the coastal
land areas and on the islands, open ocean is poorly covered with AERONET. Thus, another available network (see Sect 5.2) is used for validation of AOD retrieved over open ocean.

## 5.2 MAN

The Maritime Aerosol Network (MAN) component of AERONET provides ship-borne aerosol optical depth measurements from the Microtops II sun photometers (Smirnov et al., 2009). These data provide an alternative to observations from islands
as well as establish validation points for satellite and aerosol transport models. Since 2004, these instruments have been deployed periodically on ships of opportunity and research vessels to monitor aerosol properties over the world oceans.

The Microtops II Sun photometer is a handheld device specifically designed to measure columnar optical depth and water vapor content (Morys et al., 2001). The direct Sun measurements are acquired in five spectral channels within the spectral range 340–1020 nm. The bandwidths of the interference filters vary from 2 to 4 nm (UV channels) to 10 nm for visible and
near-infrared channels. The estimated uncertainty of the optical depth in each channel does not exceed ±0.02, which is slightly higher than the uncertainty of the AERONET field (not master) instruments as shown by Smirnov et al. (2006).

Comparison of MAN and AERONET AOD data does not show any particular bias for AERONET and MAN, although a visible cluster of points above the 1:1 line was acquired in a highly variable dust outbreak conditions west of Africa in the North Atlantic (Smirnov et al., 2011).

## 5.3 MODIS

Moderate Resolution Imaging Spectroradiometer (MODIS) was launched onboard Terra in 1999. It has a wide spectral range from 0.41μm to 14.5μm, broad swath of 2330 km, and relatively fine spatial resolution of 250 m to 1 km (Levy et al., 2013). The local equator crossing times for MODIS onboard Terra is 10:30.

In this study, the level 2 combined Dark Target and Deep Blue (DTB) AOD product (MOD04_L2) from MODIS Terra
collection C6.1 was utilized, which is characterized by good quality and better than Dark Target or Deep Blues coverage alone (Wei et al., 2019).





## 6 Validation with AERONET

The AERONET network does not cover the globe evenly. The location of AERONET stations and number of S3A collocations

per AERONET station utilized in the validation exercise are shown in Figure 1. For S3B, the number of matchups is similar

(slightly higher).

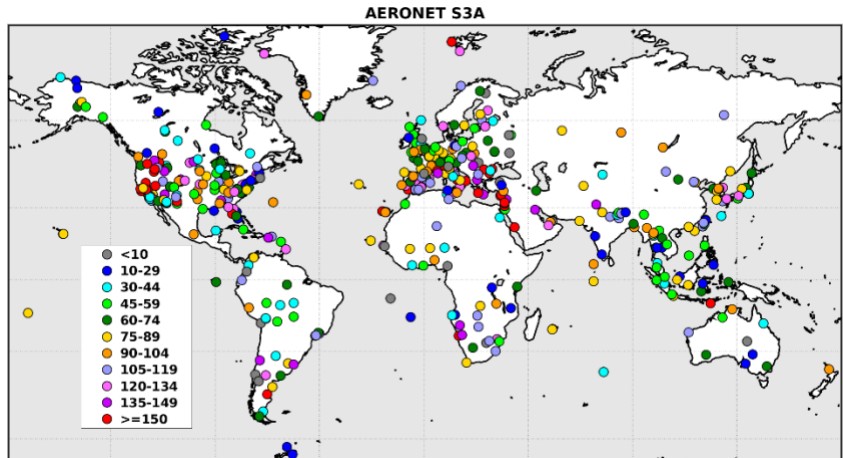

**Figure 1: Location of the AERONET stations and number of matchups with S3A, per station (see legend) for the period 2020.01-2021.09.**

In the exercise it was found that the validation results for S3A and S3B are, in general, similar (difference between results for

S3A and S3B is less than 10% of S3A AOD). In this paper, validation results for S3A are shown in figures, while validation

statistics for both S3A and S3B (shown as S3A/S3B) are summarised in tables and discussed.

### 6.1 AOD at 550nm

As shown in Figure 1, AERONET stations are not evenly distributed globally. For the study period, 01.01.2020-30.09.2021,

more than 85% of the matchups were from the NH. Thus, most of global results were strongly influenced by the results obtained

for the NH. In case validation results are similar for the globe and the NH, results for the globe are not visualised. In case of a

significant difference between the results for the globe and the NH, we show figures and discuss results for both. Validation

statistics summarised in tables include results for the globe, NH, and SH.

#### 6.1.1 Annual results

Scatter density plots for S3A SY_2 $AOD_{550}$ ($syAOD_{550}$, or syAOD) and corresponding AERONET $AOD_{550}$ ($aAOD_{550}$, or

aAOD) for all matchups available for the NH and SH, are shown in Figure 2. For most of the matchups (91 %), retrieved

syAOD is small (<0.04).

Analysis of the binned (based on aAOD) AOD was performed. For S3A, binned syAOD offset for the smallest aAOD (<0.02)

is 0.03-0.05 higher in the NH for all groups (0.11/0.09/0.016/0.08 for all/dual/singleN/singleO) compared with the results for





the SH. Offsets in the same AOD range for the S3B are slightly (up to 0.03) lower. For all matchups in the NH small syAOD

overestimation (0.07-0.12) is observed for aAOD below 1.2. For 1.2<aAOD<1.5, syAOD is slightly underestimated; for aAOD

> 1.5, syAOD is considerably underestimated. However, less than 0.5% of the matchups fit to the range of aAOD>1.5. In the

SH, binned syAOD is overestimated.

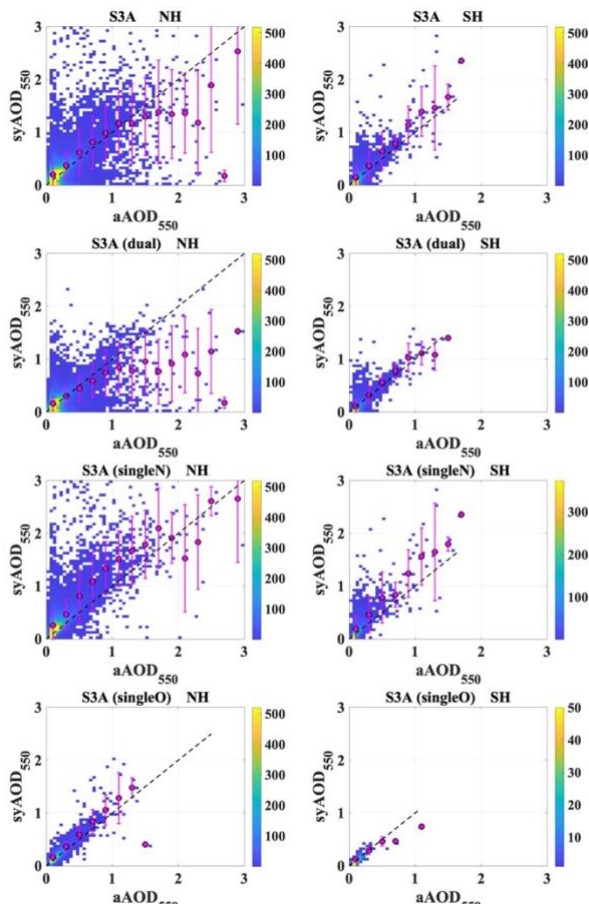

**Figure 2: Scatter density plots for S3A syAOD$_{550}$ and corresponding aAOD$_{550}$ for all, dual, singleN and singleO groups of matchups (panels top down) available over the NH (left panel) and SH (right panel).**

For 'dual' group, binned syAOD is lower than aAOD for aAOD> 0.9. For 'singleN' group, positive syAOD offset of 0.2-0.3

AOD is observed when aAOD is in the range of [0.3 1]. For 'singleO' group, offset between syAOD and aAOD is small for

aAOD<1.5. Fraction of cases with aAOD>1.5 is very small (only few matchups are available for this group).

Validation statistics (number of points, N; percentage of matchups which fit into MODIS AOD error envelope (EE) defined

as ±0.05±0.2*AOD (Remer et al., 2013); percentage of matchups which satisfy GCOS requirements of 0.03 or 10% of AOD



(GCOS, 2016); correlation coefficient (R); root mean square (rms); standard deviation, std; bias and slope defined with linear regression applied to all available matchups) for S3A and S3B products are shown in  Table 1.

**Table 1 Validation statistics (number of points, N; percentage of matchups which fit into MODIS AOD error envelope, EE, defined as ±0.05±0.2*AOD; percentage of matchups which satisfy GCOS requirements (0.03 or 10% of AOD); correlation coefficient, R; root mean square, rms; standard deviation, σ; bias and slope defined with linear regression applied to all available matchups) for S3A and S3B syAOD550 products for the globe, NH and SH for the whole period for all matchups and for three groups of matchups, defined with the processor applied.**

| group | area | N | | EE , % | | GCOS , % | | R | | rms | | std | | bias | | slope | |
|---|---|---|---|---|---|---|---|---|---|---|---|---|---|---|---|---|---|
| | | S3A | S3B | S3A | S3B | S3A | S3B | S3A | S3B | S3A | S3B | S3A | S3B | S3A | S3B | S3A | S3B |
| all | globe | 38376 | 38829 | 51,4 | 57,9 | 23,8 | 27,7 | 0,60 | 0,63 | 0,28 | 0,24 | 0,001 | 0,001 | 0,12 | 0,10 | 0,89 | 0,87 |
| | NH | 32856 | 33240 | 48,2 | 55,1 | 20,5 | 24,8 | 0,60 | 0,62 | 0,28 | 0,25 | 0,001 | 0,001 | 0,13 | 0,11 | 0,86 | 0,85 |
| | SH | 5520 | 5589 | 70,8 | 74,6 | 43,0 | 44,9 | 0,62 | 0,70 | 0,22 | 0,15 | 0,003 | 0,002 | 0,04 | 0,04 | 1,19 | 1,06 |
| dual | globe | 25098 | 25796 | 57,9 | 61,9 | 29,1 | 32,1 | 0,61 | 0,64 | 0,19 | 0,18 | 0,001 | 0,001 | 0,11 | 0,09 | 0,62 | 0,65 |
| | NH | 21430 | 21989 | 54,2 | 59,0 | 25,4 | 29,3 | 0,60 | 0,62 | 0,20 | 0,19 | 0,001 | 0,001 | 0,12 | 0,10 | 0,58 | 0,62 |
| | SH | 3668 | 3807 | 79,3 | 78,7 | 50,5 | 48,3 | 0,79 | 0,78 | 0,12 | 0,12 | 0,002 | 0,002 | 0,02 | 0,02 | 1,07 | 1,03 |
| singleN | globe | 19986 | 19936 | 37,9 | 46,2 | 14,1 | 18,1 | 0,66 | 0,67 | 0,35 | 0,30 | 0,002 | 0,002 | 0,14 | 0,12 | 1,20 | 1,13 |
| | NH | 17114 | 17084 | 35,5 | 43,6 | 11,8 | 15,4 | 0,67 | 0,67 | 0,36 | 0,31 | 0,002 | 0,002 | 0,15 | 0,13 | 1,19 | 1,12 |
| | SH | 2872 | 2852 | 51,7 | 61,8 | 27,8 | 33,9 | 0,58 | 0,62 | 0,30 | 0,19 | 0,005 | 0,003 | 0,09 | 0,07 | 1,31 | 1,11 |
| singleO | globe | 5235 | 5396 | 57,7 | 54,9 | 20,4 | 18,3 | 0,90 | 0,90 | 0,11 | 0,11 | 0,001 | 0,001 | 0,06 | 0,07 | 1,12 | 1,07 |
| | NH | 4898 | 5027 | 56,2 | 52,8 | 18,5 | 16,0 | 0,90 | 0,90 | 0,11 | 0,11 | 0,001 | 0,001 | 0,06 | 0,07 | 1,12 | 1,07 |
| | SH | 337 | 369 | 80,4 | 82,7 | 48,7 | 50,4 | 0,85 | 0,88 | 0,06 | 0,06 | 0,003 | 0,002 | 0,05 | 0,03 | 0,83 | 1,07 |


For all matchups, validation statistics are slightly better for S3B ( Table 1): more matchups fit to the EE, R is higher, rms, standard deviation and total bias are lower.

A difference in the algorithm performance in the NH and SH can be seen. The EE and GCOS fractions are considerably higher for the SH, but R and rms are only slightly improved.

In addition to the statistics shown in Table 1, we performed respective analysis for limited AOD ranges. For aAOD<1.5, syAOD validation statistics are slightly better than statistics for all aAOD ranges: bias is close to 0.1, slope is close to 1 for both S3A and S3B AOD products in the NH. For aAOD>1.5, bias is ca. 1.3 in the NH (where N is 127/125 for S3A/S3B, respectively). In the SH matchups available for S3B product are located close to the 1:1 line, however the number of matchups with aAOD>1.5 is small (N is 3/2) to calculate validation statistics.

Group analysis reveals that most of the low biased syAOD outliers were retrieved with the dual processor (Figure 2), while most of the high biased syAOD outliers were retrieved with the single processor applied to the nadir view (group singleN). Total bias is smaller for dual group globally, and in both NH and SH ( Table 1).  For aAOD<1.5, syAOD bias is close to 0 for the dual group; for the single group bias is higher than for all matchups and increasing with aAOD. Validation statistics are,



in general, better in the SH (except for R for all the single groups). As for all matchups, validation statistics are slightly better

for S3B.

Global difference (dAOD, represented with the median bias and bias standard deviation) between Sy_2 and AERONET monthly AOD for selected aAOD bins are shown in Figure 3 for all aerosol types (including background (aAOD ≤ 0.2) AOD), fine-dominated and coarse-dominated AOD. Globally, background AOD (64% from all matchups) is overestimated by 0.04-0.06. Overestimation of fine-dominated matchups is increasing from 0.07 to 0.15 in the AOD range of 0.2-1.2 (34% of

matchups). Overestimation for coarse-dominated matchups is about 0.05 for aAOD<0.7; for aAOD of 0.7-0.9, an overestimation for coarse-dominated matchups is within the GCOS requirements of ±0.03 dAOD. For aAOD>1.2 (less than 2% of matchups), dAOD is varying in the sign and in the amplitude; however, the number of matchups in AOD bind >1.2 is low and results are thus unstable. Fine-dominated matchups dominate over coarse-dominated. Fractions of fine-dominated matchups per bin is 60-70% for aAOD in the range of 0.2-0.9 and more than 70% for aAOD>0.9. Thus, binned offsets for all

matchups follow closely offsets for fine-dominated matchups.

In the NH, the AOD offset for the background matchups is slightly higher (~0.07); in the SH the offset is lower (<0.02). Binned offsets for the fine-dominated and coarse-dominated matchups in the NH are similar from those for the globe. In the SH, offsets of syAOD are higher for aAOD>0.4, where the number of the matchups per bin is limited (<50).


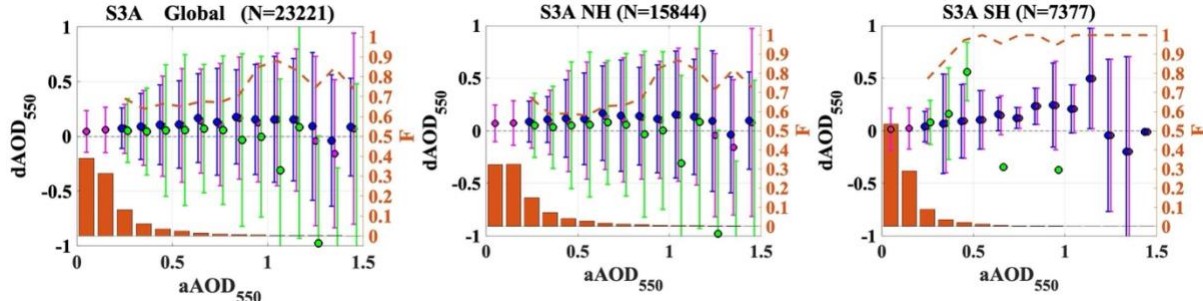

**Figure 3: Global, as well as for the NH and SH (left to right), difference (dAOD$_{550}$) between syAOD and aAOD for selected aAOD bins: median bias (circles) and bias standard deviation (error bars) for all and background (aAOD ≤ 0.2) AOD types (purple), aerosol fine-dominated AOD (blue) and coarse-dominated AOD (green). The fraction (F) of points in each bin from the total number of**
**matchups is represented by orange bars. The fraction of fine-dominated matchups in each bin is shown as orange dashed-line.**

### 6.1.2    Monthly and seasonal results

For the period 01.2020-09.2021, monthly (Jan, Feb, Mar, etc.), seasonal (DJF, MAM, JJA, SON) and annual (Year) validation results for S3A and S3B syAOD for the globe, NH and SH are summarised as time series plots in Figure 4.

Correlation coefficient R is of sinusoidal shape for monthly statistics with two maxima for both S3A and S3B in the NH ($R_{NH}$).

Peaks in $R_{NH}$ are observed in Feb-Apr (~0.65-0.70) and Aug-Sep. For both S3A and S3B, first clear minimum (0.45-0.47) is observed in Jun; second minimum is observed in Nov-Dec-Jan for S3A (0.48-0.52) and S3B (0.55-0.58). In the SH, correlation





coefficient $R_{SH}$ varies strongly along the year. A clear peak (0.8-0.9) for both S3A and S3B is observed in Jun-Oct. In Dec-May, $R_{SH}$ is between 0.40 and 0.65 for S3B. For S3A, $R_{SH}<0.4$ in Dec-Mar.

Rms$_{NH}$ is within 0.25-0.32 for both S3A and S3B, with minimum in Oct-Jan and maximum in Mar-May. In the SH, rms$_{SH}$ for
S3B is 0.15-0.2 in Dec-May and 0.09-0.14 in the other months. It is higher for S3A, reaching values of 0.25-0.35 in Jan-Feb and Nov-Dec.

Bias$_{NH}$ varies from 0.06 to 0.14 in monthly statistics. For S3B, bias is 0.01-0.35 lower than for S3A in all months, except for April. Bias$_{SH}$ is lower; it varies from 0.01 to 0.08 in monthly statistics.

Slope$_{NH}$ is below 1 in Aug-Jan and above 1 in Apr-May. Slope$_{SH}$ is in general above 1 and is higher than slope$_{NH}$ in Jun. Slope
for S3B is lower than for S3A.

The fraction of matchups in the EE reflects well the difference between the NH and SH and between S3A and S3B. Fraction of the matchups in EE in the SH is considerably higher than in the NH (up to 25% difference) in Feb-Sep, with maximum of 75-82% in the SH in May-Aug for both instruments. EE is, in general, higher for S3B with the offset up to 15% in the NH.

Seasonal statistics are as following. $R_{NH}$ is 0.55-0.65 for all seasons for both S3A and S3B. $R_{SH}$ is higher in JJA (0.65 and 0.83
for S3A and S3B, respectively) and SON (0.8 and 0.84) and lower in DJF (0.32 and 0.5) and MAM (0.45 and 0.47). Rms$_{NH}$ is higher in MAM (0.28 and 0.32 for S3A and S3B, respectively) and lower (0.2-0.27) in other seasons. Rms$_{SH}$ has opposite behaviour, with minima in MAM and JJA for S3A and in JJA and SON for S3B. Sigma is higher for S3A in the SH. The number of points in the EE is clearly higher for S3B and in the SH, especially in MAM, JJA and SON.

Annual results are summarised in Sect. 6.1.1.

As a short summary, syAOD550 validation results show slightly better quality of S3B syAOD$_{550}$; retrieval algorithm produces better results in the SH.



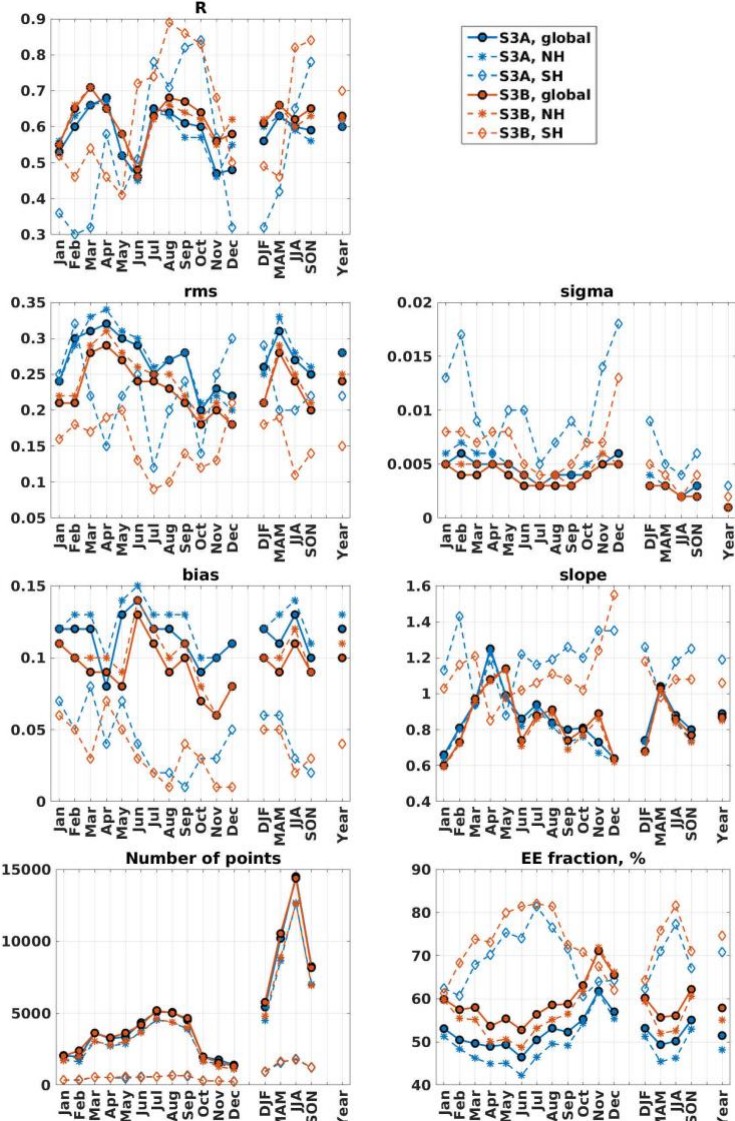

**Figure 4: Validation statistics for syAOD$_{550}$ aggregated monthly (Jan, Feb, …, Dec), seasonally (DJF, MAM, JJA, SON) and yearly (Year) shown as time series for S3A and S3B for the globe, NH and SH.**

### 6.1.3 Regional performance

There are noticeable regional differences in the performance of the retrieval algorithm, which depend on, e.g., AOD load and AOD types (composition and optical properties), as well as on the properties of underlying surfaces. Retrieval quality (accuracy, precision and coverage) varies considerably as a function of these conditions, as well as whether a retrieval is performed over land or over ocean.

Following Sogacheva et al. (2020), we inter-compare validation results over 15 regions (as defined in Table 2) that seem likely to represent a sufficient variety of aerosol and surface conditions. These are shown in Figure 5 and include 11 land regions,

two ocean regions and one heavily mixed region. The land regions represent Europe (denoted by Eur), Boreal (Bor), northern, eastern, and western Asia (AsN, AsE and AsW, respectively), Australia (Aus), northern and southern Africa (AfN and AfS), 395 South America (AmS), and eastern and western Northern America (NAE and NAW). South-Eastern China (ChinaSE), with is part of the AsE, is considered separately. The Atlantic Ocean is represented as two ocean regions, one characterised by Saharan dust outflow over the central Atlantic (AOd) and a second that includes burning outflow over the southern Atlantic (AOb). The mixed region over Indonesia (Ind) includes both land and ocean.

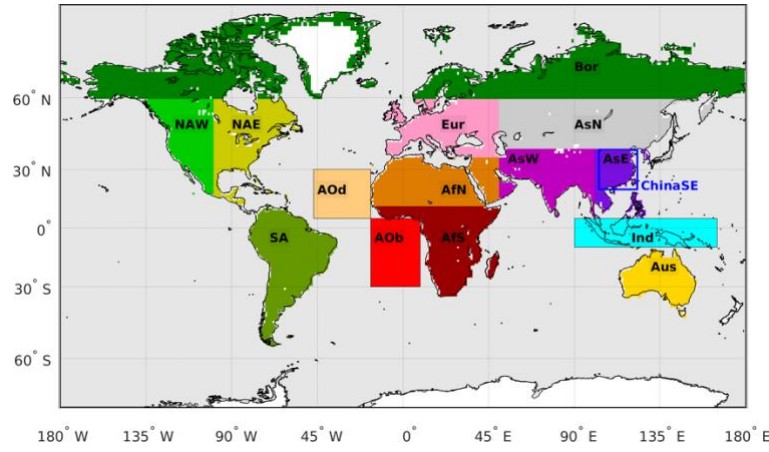


**Figure 5: Land and ocean regions defined for this study (as in Sogacheva et al., 2020): Europe (Eur), Boreal (Bor), northern Asia (AsN), eastern Asia (AsE), western Asia (AsW), Australia (Aus), northern Africa (AfN), southern Africa (AfS), South America (SA), eastern North America (NAE), western North America (NAW), Indonesia (Ind), Atlantic Ocean dust outbreak (AOd), Atlantic Ocean biomass burning outbreak (AOb). In addition, Southeast China (ChinaSE), which is part of the AsE region, marked with a**
**blue frame, is considered separately. Land, ocean and global AOD were also considered.**

High diversity in the validation results was observed between the selected regions (Figure 6, Table 2). Highest correlation (0.94) was found in AOb region (total number of matchups is low (22) in that region). For ChinaSE, AsN, AsE, AOd, Aus, NAE, correlation coefficient R was in the range 0.6-0.8, which was higher than that for the globe. For Eur and Ind, R <0.4. Bias did not change much between the regions. Bias between binned syAOD and aAOD was positive in Asia, Bor and SA 410 regions for aAOD< ~1.2; bias calculated with linear regression was higher for those regions. The amount of syAOD outliers, defined as |syAOD-aAOD| >0.5, varied among the regions. In Eur, positive syAOD outliers were observed for aAOD<0.3. For Asian and Bor regions, syAOD outliers were observed mostly for aAOD in the range of [0.2 1.2]. More negative syAOD outliers were observed in the NAW region.

Among the land regions, the fraction of the pixels in EE was highest in Aus (81,6%), lowest in Bor and SA (<30%); for other 415 land regions fraction of the pixels in EE was in the 30%-60% interval. Over ocean, in AOb and AOd (with only 22 matchups) areas, fraction of the pixels in EE was high (67,8% and 95,5%, respectively).

The fraction of syAOD pixels which satisfy GCOS requirements was low (<31%) for all regions, except for Aus (54,5%) and AOb (68,2%).



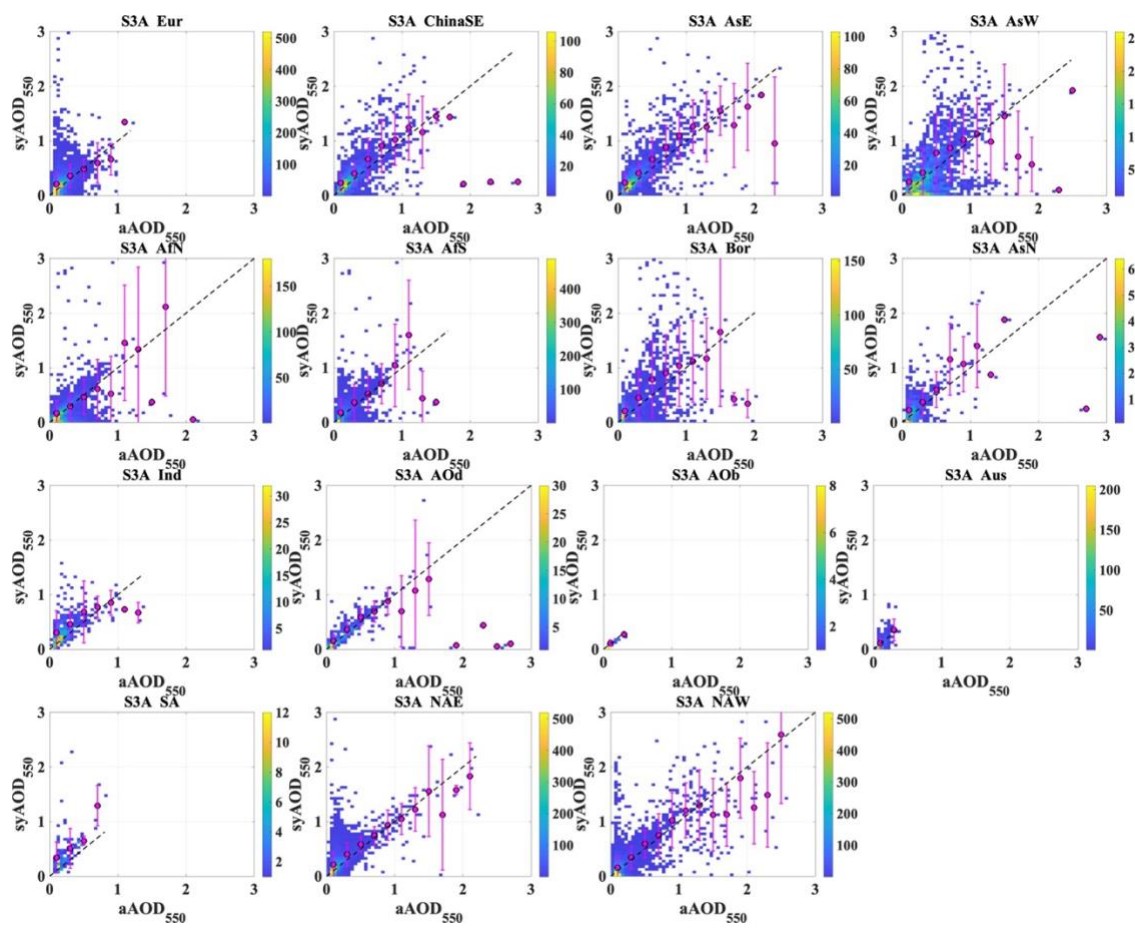


**Figure 6: For S3A, syAOD and aAOD scatter density plots for selected regions (as defined in Figure 5).**





**Table 2: For S3A, regional statistics of comparison between syAOD$_{550}$ and AERONET: number of points, N; percentage of matchups which fit into MODIS AOD error envelope, EE; percentage of matchups which satisfy GCOS requirements; correlation coefficient, R; root mean square, rms; standard deviation, sigma; bias and slope defined with linear regression applied to all available matchups.**

| Region | coverage | N | EE, % | GCOS, % | R | rms | sigma | bias | slope |
|---|---|---|---|---|---|---|---|---|---|
| Eur | 36°N-60°N, 15°E-50°W | 9226 | 50,4 | 19,0 | 0,36 | 0,24 | 0,002 | 0,13 | 0,84 |
| Bor | 60°N-85°N, 180°W-180°E | 1737 | 27,9 | 12,4 | 0,55 | 0,47 | 0,011 | 0,17 | 0,96 |
| AsN | 40°N-60°N, 50°°E-165E | 784 | 43,9 | 18,9 | 0,64 | 0,28 | 0,009 | 0,15 | 0,81 |
| AsE | 5°N-40°N, 100°E-130°E | 2366 | 49,7 | 19,4 | 0,73 | 0,29 | 0,005 | 0,14 | 0,96 |
| ChinaSE | 5°N-30°N, 103°E-135°E | 2298 | 46,9 | 17,7 | 0,67 | 0,29 | 0,005 | 0,14 | 0,93 |
| AsW | 5°N-40°N, 50°E-100°E | 1780 | 32,0 | 13,5 | 0,43 | 0,57 | 0,013 | 0,23 | 0,82 |
| Aus | 10°S-45°S, 110°E-155°E | 626 | 81,6 | 54,5 | 0,64 | 0,09 | 0,003 | 0,01 | 1,30 |
| AfN | 12°N-36°N, 17°S-50°N | 3049 | 54,2 | 20,2 | 0,44 | 0,30 | 0,005 | 0,10 | 0,71 |
| AfS | 35°S-12°N, 9°E-50°E | 3298 | 55,4 | 30,4 | 0,47 | 0,29 | 0,005 | 0,09 | 0,96 |
| SA | 0°N-13°N,47°W-82°W | 134 | 22,4 | 6,7 | 0,54 | 0,38 | 0,026 | 0,16 | 1,39 |
| NAE | 13°N-60°N, 55°W-100°W | 6124 | 46,6 | 20,3 | 0,56 | 0,25 | 0,003 | 0,13 | 0,91 |
| NAW | 13°N-60°N, 100°W-135°W | 5834 | 55,8 | 30,9 | 0,67 | 0,26 | 0,003 | 0,10 | 0,80 |
| Ind | 10°S-5°N, 90°E-165°E | 604 | 40,6 | 17,4 | 0,33 | 0,42 | 0,015 | 0,23 | 0,74 |
| AOd | 5°N-30°N, 17°W-47°W | 369 | 67,8 | 29,8 | 0,64 | 0,30 | 0,015 | 0,18 | 0,56 |
| AOb | 30°S-5°N, 17°W-9°E | 22 | 95,5 | 68,2 | 0,94 | 0,03 | 0,005 | 0,03 | 0,92 |


Regional differences between syAOD and aAOD for all aerosol types (including background (aAOD ≤ 0.2) AOD), fine-dominated and coarse-dominated AOD for selected aAOD bins are shown in Figure 7. A general tendency towards positive SY_2 AOD offsets is observed for most regions under the background conditions. Offsets are higher (up to 0.15) in Ind and SA and lower (below 0.04) in AfN, AfS and AOd. The behavior of the fine-dominated offset is similar for most of the regions

(ChinaSE, AfN, AfS, Ind) with gradual increase in the aAOD range of ca 0.7-1.1. Coarse-dominated offset over Eur is underestimated by up to 0.18 for aAOD of 0.6-0.8. Over China, coarse-dominated offset is slightly overestimated at aAOD<0.7 and underestimated at aAOD>1. Over bright surface with contribution of dust aerosols (AfN), all groups show a good agreement with aAOD for aAOD<0.7. For aAOD>0.7, syAOD for coarse-contaminated matchups is considerably underestimated. Similar offsets are observed in NAE region, where 70-90% of matchups are characterized with fine-dominated

aerosols. In possible biomass burning region (AfS), an underestimation of syAOD for coarse-dominated matchups gradually increases for aAOD>0.3 reaching -0.9 at aAOD close to 1. Over Ind, dAOD is positive for aAOD <0.5. Over ocean, with possible contamination of Saharan dust (AOd), offsets are constantly slightly (up to 0.1) positive for all groups at aAOD<1.



Figure 7: Regional (for Eur, ChinaSE, AfN, AfS, Ind, AOd, SA, NAE) difference (dAOD$_{550}$) between syAOD and aAOD for selected
aAOD bins: median bias (circles) and bias standard deviation (error bars) for all and background (aAOD $\leq$ 0.2) AOD types (purple),
aerosol fine-dominated AOD (blue) and coarse-dominated AOD (green). The fraction (F) of points in each bin from the total number
of matchups is represented by orange bars. The fraction of fine-dominated matchups in each bin is shown as orange dashed-line.
Results for other regions are in the Supplement (Figure S 5)



### 6.1.4 Analysis of syAOD relative offsets

syAOD relative offset, or dAOD$_{rel}$, was defined as in eq.1:

dAOD$_{rel}$ = (syAOD-aAOD)/aAOD  (eq.1)

syAOD offset analysis was performed for matchups which did not satisfy the GCOS requirements of |syAOD-aAOD|<0.03 or |syAOD-aAOD|<0.1*aAOD (GCOS, 2016).

#### 6.1.4.1 Latitude dependence of the syAOD relative offset

In Figure 8 we show a density scatter plot for the latitude dependence of the relative offset of the AOD for all, dual, singleN and singleO groups of pixels for S3A and S3B products. Colour indicates the fraction of the points with corresponding dAODrel from the total number of points within the latitude bin. As an example, for the latitude in [-30 -20], which means 20°S-30°S, dAODrel was between -0.5 and -1 for ca 38% (colorbar) of matchups. Magenta line shows the number of matchups in x-axis bin.

In the NH, dAODrel was mostly positive (syAOD was higher than aAOD). In the SH, dAODrel was mostly positive in 30°S-60°S and mostly negative in 10°S-30°S, except for the singleN group, where dAODrel was mostly positive. In both NH and SH, dAODrel was increasing towards the poles. This increase was more pronounced for the singleO group of pixels, but also visible in the dual group.

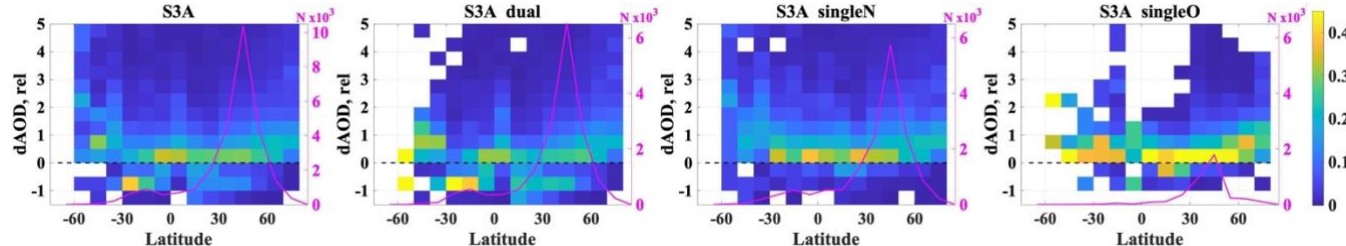


**Figure 8: For S3A, density scatter plot for latitude dependence of the syAOD relative offset for 'all', 'dual' and 'singleN' groups of pixels (vertical panels from left to right, respectively). Colour indicates the fraction of the points with corresponding dAODrel interval from the total number of points within the latitude bin. Magenta line shows the total number of the matchups in the corresponding latitude bin.**

#### 6.1.4.2 Dependence of syAOD relative offset on surface reflectance.

The directional surface reflectance (SR) retrieved with the SYNERGY algorithm is provided in the SY_2_AOD product.

In Figure 9 we show a density scatter plot for the dependence of the relative offset of the AOD on the retrieved SR for the globe, NH and SH. Colour indicates the fraction of the points with corresponding dAODrel from the total number of points within the surface reflectance bin.

For all matchups (not shown here), as well as for the dual group (globally, as on Figure 9, as well as over the NH and SH), footprints for the dAODrel dependence on the SR are similar. For SR< 0.05 and SR>0.35, dAODrel indicates that syAOD is mostly overestimated. In specified ranges, dAODrel is increasing towards outer edges. For the SR in the range of 0.05-0.35,





syAOD is mostly underestimated. Underestimation is more pronounced when syAOD is retrieved with the dual processor. For the singleO group, syAOD is mostly overestimated in all SR ranges.


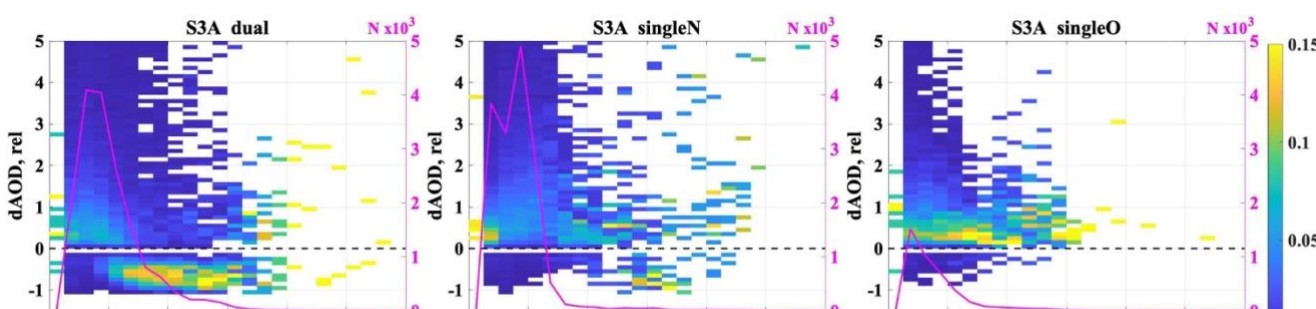

**Figure 9: For S3A, syAOD matchups with AERONET which do not satisfy GCOS requirements, scatter density plot for the dependence of the syAOD relative offset of retrieved surface reflectance for 'all', 'dual' and 'singleN' groups of pixels (vertical panels from left to right, respectively). Colour indicates the fraction of the points with corresponding dAODrel interval from the total number of points within the surface reflectance bin. Magenta line shows the total number of the matchups in the corresponding surface reflectance bin.**


### 6.1.4.3 Dependence of the AOD relative offset on solar and satellite geometry

In Figure 10 we show the dependence of the syAOD relative offsets on the OLCI geometry (relative azimuth (Raz), satellite zenith angle (SatZA) and sun (or solar) zenith angle (SunZA) provided in the SY_2_AOD product, North and Heckel, 2019)

for the NH and SH. Colour indicates the fraction of the points with corresponding dAODrel interval in the Raz, SatZA, or SunZA bins.

In the NH, positive dAODrel is increasing for Raz in [50° 80°] and in [100° 140°]. In the SH, we see the similar dependence of dAODrel for Raz in [50° 80°]. For Raz>90°, positive dAODrel is increasing with Raz increase from 150° to 180°; negative dAODrel of [-1 -0.5] is observed more often than positive [0 0.5] dAODrel.

No significant dependence of dAODrel on the SatZA was observed. However, a greater number of negative dAODrel is clearly seen in the SH.

In the NH, dAODrel is slightly positive (0-0.5), in all range of SunZA, except for the most extreme values. For SunZA>80°, the percentage of higher positive dAODrel (0.5-1) increases, while for SunZA<30° the percentage of higher negative dAODrel rises. In the SH, similar dependence was observed, except for SunZA in the range of 50°-65°, where dAODrel is mainly

negative.



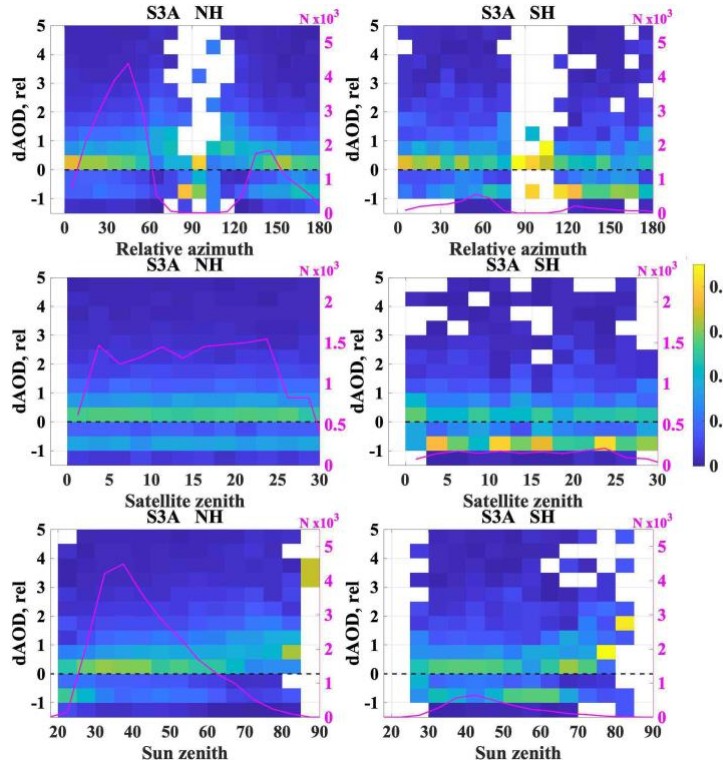

**Figure 10: For S3A, syAOD matchups with AERONET which do not satisfy GCOS requirements, the dependence of the AOD relative offsets on relative azimuth (upper panel), satellite zenith angle (middle panel) and sun zenith angle (lower panel) for the NH (left panel) and SH (right panel) for all pixels. Colour indicates the fraction of the points with corresponding dAODrel from the total number of points within the x-axis bin. Magenta line shows the total number of the matchups in the corresponding x-axis bin.**

### 6.1.5 Linear regression considering provided syAOD uncertainties

Linear fitting for combinations of $syAOD_{550}$ and $aAOD_{550}$ collocations has been performed with a consideration of the $syAOD_{550}$ and $aAOD_{550}$ uncertainties (https://se.mathworks.com/help/stats/linearmodel.predict.html, last access 08.03.2022). For $syAOD_{550}$, pixel-level uncertainties are provided in the SY_2_AOD product. For $aAOD_{550}$, uncertainty of 0.01 has been considered (Eck et al., 1999).

In Figure 11 we show linear fit for $syAOD_{550}$ and $aAOD_{550}$ without (green line) and with (red line) considering the provided $syAOD_{550}$ uncertainties for all matchups and groups of pixels retrieved with different (dual, single) approaches. Corresponding statistics (bias, slope) are provided in Table 3.





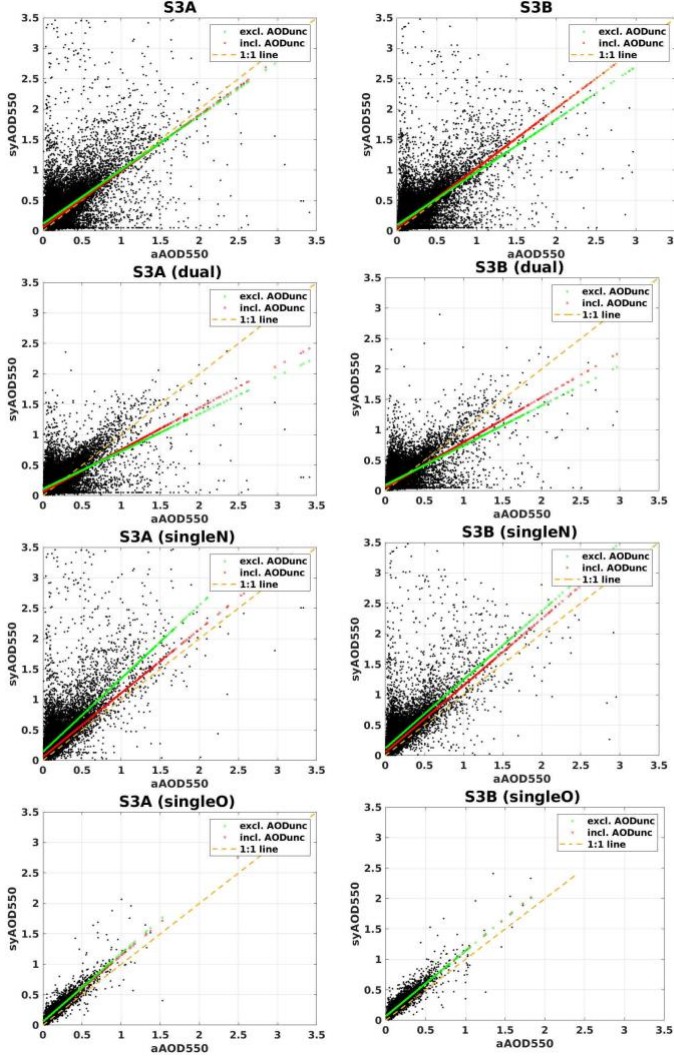

**Figure 11: For S3A (left panel) and S3B (right panel), linear fit without (green dotted line) and with (red dotted line) considering syAOD$_{550}$ and aAOD$_{550}$ uncertainties. Dashed orange line is 1:1 line.**

For both S3A and S3B, for all groups of matchups, bias and slope for the linear regression fits applied to the whole AOD range were improved when the syAOD and aAOD uncertainties were considered. Bias was lowered roughly by 50%. Slope was improved by 10-15%. Improvements were smaller for singleO group of matchups (retrievals over ocean), for which the

syAOD uncertainties are smallest (Sect. 6.2).





**Table 3: Validation statistics (bias, slope) for groups of pixels for syAOD$_{550}$ and aAOD$_{550}$ without and with consideration of AOD uncertainties**

| | Consider AOD uncertainty | bias | | slope | |
|---|---|---|---|---|---|
| | | S3A | S3B | S3A | S3B |
| **all** | no | 0.12 ± 0.002 | 0.10 ± 0.001 | 0.89 ± 0.006 | 0.86 ± 0.003 |
| | yes | 0.06 ± 0.001 | 0.05 ± 0.001 | 0.92 ± 0.003 | 0.99 ± 0.003 |
| **dual** | no | 0.11 ± 0.001 | 0.09 ± 0.001 | 0.62 ± 0.005 | 0.65 ± 0.005 |
| | yes | 0.06 ± 0.001 | 0.06 ± 0.001 | 0.69 ± 0.005 | 0.74 ± 0.005 |
| **singleN** | no | 0.14 ± 0.003 | 0.12 ± 0.002 | 1.20 ± 0.010 | 1.13 ± 0.009 |
| | yes | 0.07 ± 0.001 | 0.05 ± 0.001 | 1.05 ± 0.004 | 1.11 ± 0.004 |
| **singleO** | no | 0.06 ± 0.002 | 0.07 ± 0.002 | 1.22 ± 0.007 | 1.07 ± 0.007 |
| | yes | 0.06 ± 0.001 | 0.07 ± 0.001 | 1.08 ± 0.007 | 1.07 ± 0.007 |

### 6.1.6   AOD at other than 550nm wavelengths

Scatter plots for SY_2 AOD$_{440}$, AOD$_{670}$, AOD$_{865}$, and AOD$_{1600}$ are shown in Figure 12. Clear tendencies in validation statistics were observed when comparing validation results from shorter (440nm) to longer (1600nm) wavelengths. Though the correlation coefficient is decreasing (0.65/0.55/0.50/0.40 for 440/670/865/1600 nm respectively), the offset (0.15/0.1/0.07/0.05/) and rms (0.33/0.23/0.18/0.16) are also decreasing.

Validation statistics for all wavelengths are slightly worse for the NH than global validation statistics (Table 4); validation
statistics for the SH are considerably better than for the NH (except for R for 1600nm wavelength).



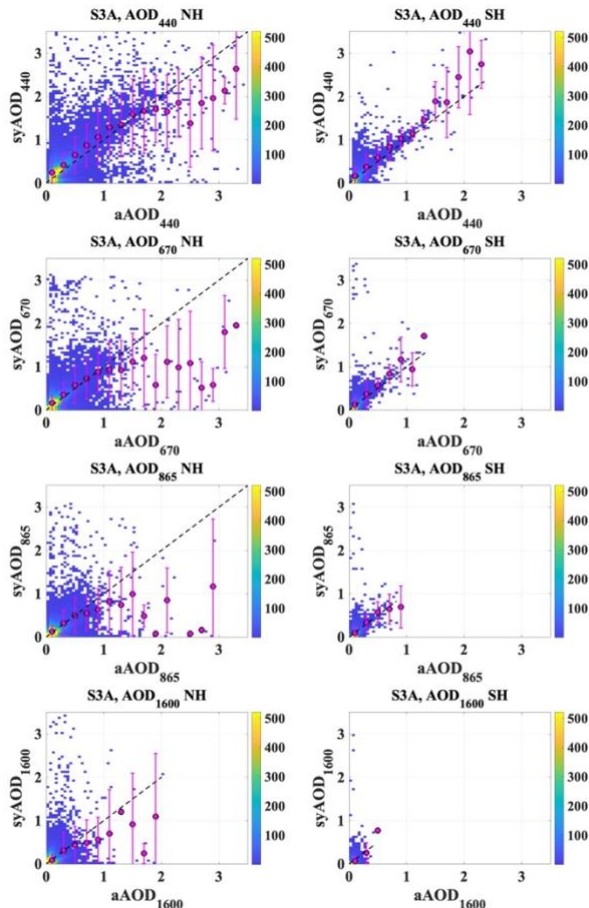

**Figure 12: Scatter plots for SY_2 AOD$_{440}$, AOD$_{670}$, AOD$_{870}$, and AOD$_{1600}$ (panels top down) for the NH and SH (left and right panels, respectively).**

syAOD$_{440}$ is overestimated for all aerosol types (Figure 13). syAOD$_{670}$ for fine-dominated matchups is in a good agreement

with aAOD$_{670}$ for aAOD$_{670}$<1. Similar tendency, though for narrower aAOD ranges (aAOD$_{870}$<0.5 and aAOD$_{1600}$<0.3), is

observed for syAOD$_{865}$ and syAOD$_{1600}$. For all wave lengths, coarse-dominated syAOD is retrieved accurately for aAOD

below ca. 0.4; above 0.4 syAOD is underestimated and offset between syAOD and aAOD is increasing with increasing aAOD.



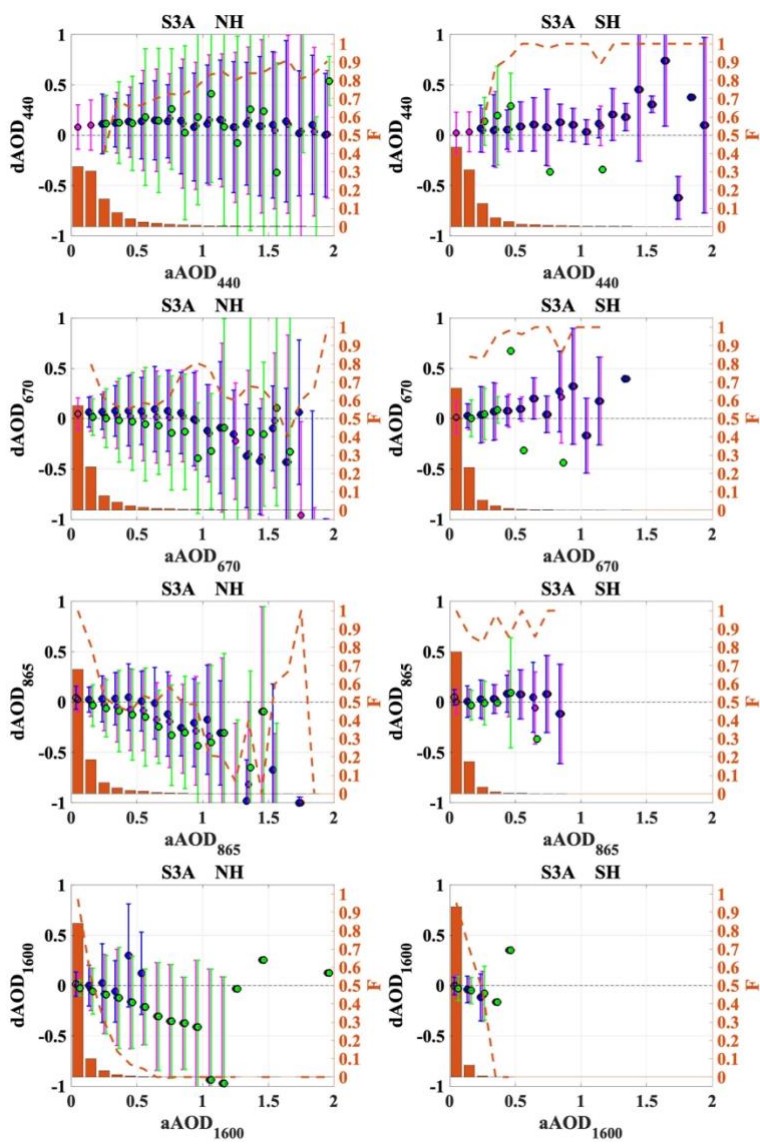

Figure 13: for the NH and SH (left to right), for different wavelengths (top down: 440, 670, 865, 1600 nm) difference
(dAOD$_{550}$) between syAOD and aAOD for selected aAOD bins: median bias (circles) and bias standard deviation (error bars)
for all (incl. background, aAOD$_{550}$ ≤ 0.2) AOD types (purple), aerosol fine-dominated AOD (blue) and coarse-dominated
(green) AOD. The fraction (F) of fine-dominated matchups from the total number of matchups in each bin is represented by
orange bars. The fraction of fine-dominated matchups in each bin is shown as orange dashed-line.





**Table 4: For S3A, for different wavelength regional validation statistics: number of points, N; percentage of matchups which fit into MODIS AOD error envelope for AOD$_{550}$, EE; percentage of matchups which satisfy GCOS requirements; correlation coefficient, R; root mean square, rms; standard deviation, sigma; bias and slope defined with linear regression applied to all available matchups.**

| wavelength, nm | Region | N | EE,% | R | rms | sigma | bias | slope |
|---|---|---|---|---|---|---|---|---|
| 440 | globe | 38119 | 41,3 | 0,65 | 0,33 | 0,002 | 0,15 | 0,90 |
| | NH | 32631 | 37,4 | 0,64 | 0,35 | 0,002 | 0,17 | 0,88 |
| | SH | 5488 | 64,5 | 0,72 | 0,24 | 0,003 | 0,06 | 1,13 |
| 670 | globe | 36955 | 58,5 | 0,55 | 0,23 | 0,001 | 0,10 | 0,77 |
| | NH | 31595 | 55,6 | 0,55 | 0,24 | 0,001 | 0,11 | 0,75 |
| | SH | 5360 | 75,7 | 0,59 | 0,18 | 0,002 | 0,04 | 1,14 |
| 875 | globe | 37273 | 69,1 | 0,50 | 0,18 | 0,001 | 0,07 | 0,71 |
| | NH | 31887 | 66,9 | 0,50 | 0,19 | 0,001 | 0,08 | 0,70 |
| | SH | 5386 | 82,3 | 0,50 | 0,13 | 0,002 | 0,02 | 1,06 |
| 1600 | globe | 29225 | 72,9 | 0,40 | 0,16 | 0,001 | 0,05 | 0,77 |
| | NH | 24950 | 71,1 | 0,40 | 0,16 | 0,001 | 0,05 | 0,76 |
| | SH | 4275 | 83,4 | 0,24 | 0,11 | 0,002 | 0,03 | 0,74 |

## 6.2    AOD uncertainties

The concept for validation of the AOD uncertainties applied in the current study follows the validation strategy suggested by Sayer et al. (2013, 2020) with consideration of the validation practice further developed in the Aerosol_cci+ project (Product Validation and Intercomparison Report, https://climate.esa.int/media/documents/Aerosol_cci_PVIR_v1.2_final.pdf, last access: 25.02.2022).

Definitions for uncertainties in the current evaluation of uncertainties are as following:

- Prognostic (per-retrieval) uncertainties (PU) for SY_2_AOD product are provided at 440, 550, 670, 865, 1600 and 2250 nm wavelengths.

- Expected discrepancy (ED) is an uncertainty variable which accounts for the PU and the accuracy of the ground-based (AERONET) data (AU), as defined by Sayer et al. (2020) in eq.2:

$$ED = \sqrt{PU^2 + AU^2} \text{ (eq.2)}$$

According to Giles et al. (2019), AU = 0.01.

- AOD error (AODerror) is a difference between satellite product AOD (syAOD) and AERONET AOD (aAOD); AOD absolute error (absAODerror) is an absolute value for AODerror.

Mean-bias correction has been performed for the error distributions in some of the subsequent analysis, since the concept of standard uncertainties requires bias-free error distributions which can be interpreted as absence of remaining systematic and quantifiable biases (https://climate.esa.int/media/documents/Aerosol_cci_PVIR_v1.2_final.pdf, last access: 25.02.2022).

If wavelength is not specifically mentioned, all variables in Section 6.2 are referring to the wavelength of 550nm.





Analysis of the distribution of the uncertainties has been performed for the whole S3A and S3B SY_2_AOD product, as well as for groups of pixels retrieved with different retrieval approaches (dual, singleN, singleO). Results for S3A and S3B are
similar; only results for S3A are shown and discussed.

### 6.2.1 χ2 test for evaluation of the prognostic uncertainties

The goodness of the predicted uncertainties was estimated with the χ2 test, as in eq.3

$$\chi2 = \frac{1}{N-1}\sum_{i=1}^{N}\bar{\delta_i} \quad \text{(eq.3)},$$

where individual weighted deviation $\bar{\delta_i}$ is described in eq.3.

$$\bar{\delta_i} = \frac{(syAOD_i - aAOD_i - mean(syAOD - aAOD))^2}{PU_i^2 + AU^2} \quad \text{(eq.4)}$$

If χ2 ~1, prognostic uncertainties describe well the AODerror. If χ2 >>1, PU are strongly underestimated; if χ2 <<1, PU are strongly overestimated. χ2 was calculated for the whole dataset and for different AOD bins to reveal if the goodness of the PU uncertainties is AOD dependent.

For the whole dataset, χ2 =3.1, which means that PU are slightly underestimated. For the binned AOD, χ2 is varying strongly
(Figure 14). For aAOD<0.4, which is ca 90% of all values, χ2 fits into the interval [1.8 3.2]. Thus, for most of the matchups, PU is only slightly underestimated. For AOD>0.4 PU underestimation is more pronounced.

No significant dependence of $\bar{\delta_i}$ on AODerror or surface reflectance provided in the SY_2_AOD product has been revealed (Figure S 6, Supplement).

Though the number of the matchups in the whole dataset is high (which provides the confidence to χ2 test results), it was
noticed that high $\bar{\delta_i}$ (up to 155) exists, which may bias the evaluation of the PU with χ2. To remove possible contribution of the outliers on the χ2 test results, cases with $\bar{\delta_i} > 10$ (which are less than 5% of the total number of matchups) were removed from the analysis.

For the dataset with the removed outliers, χ2 =1.2, which means that PU describe well the AODerror.

Influence of $\bar{\delta_i}$ outliers is more pronounced for AOD bins, where the number of matchups per bin is lower and thus the
contribution of the outlies to the results is more expected. If $\bar{\delta_i}$ outliers are removed from the binned analysis, χ2 fits to the range [1 1.45] for AOD<0.4 (Figure 14).





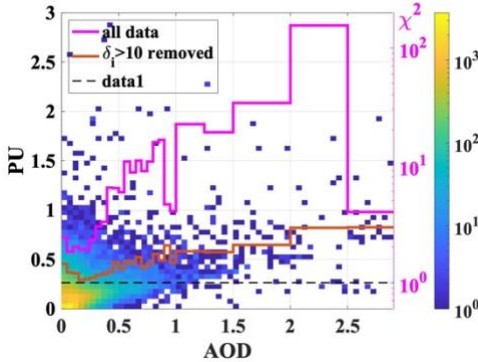

**Figure 14: χ2 for binned aAOD for all available matchups (magenta line) and after the outliers of the individual weighted deviations ($\bar{\delta}_i$>10) are removed (red line). Density scatter plot for PU and syAOD.**

### 6.2.2 Evaluation of prognostic uncertainties with absolute AOD error

To qualitatively illustrate an accuracy of prognostic uncertainties, we show in Figure 15 the comparison between the PU, AOD error distribution, and theoretical Gaussian distribution (with a mean of 0 and standard deviation of the AODerror). PU distribution shows a double peak, (first peak is at ca. 0.02-0.04 for all groups; the second peak in a range of 0.12-0.18, for different groups). For singleN, two peaks are located close to each other. Mean PU for dual group is higher; std is higher for singleN group. AOD error distributions are Gauss-like with partly some asymmetry in positive AODerror direction.

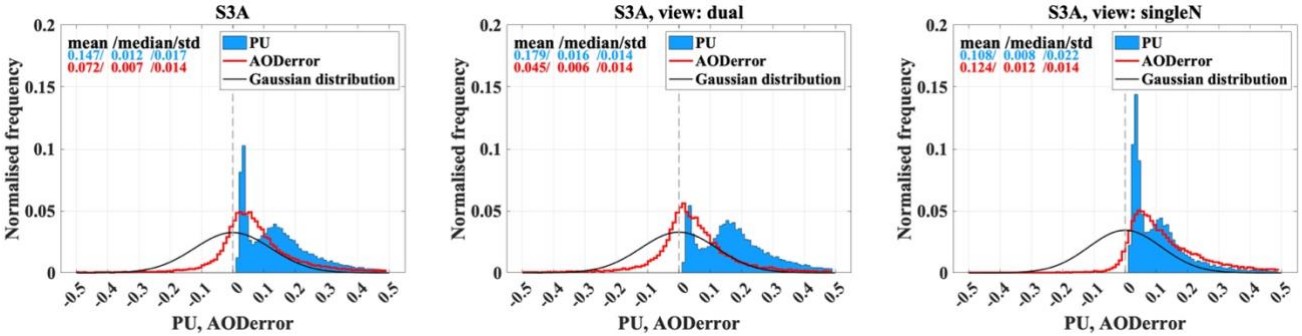

**Figure 15: Comparison between PU, AOD error distribution and theoretical Gaussian distribution for the whole product (left panel), dual- (middle panel) and singleN (right panel) groups of matchups.**

### 6.2.3 Evaluation of expected discrepancy and absolute AOD error

ED is calculated for each pixel by combining PU and AERONET uncertainties, as in eq.2.

For a quantitative validation, we follow (with some modifications) a new approach developed by ESA Aerosol CCI (https://climate.esa.int/media/documents/Aerosol_cci_PVIR_v1.2_final.pdf, last access: 25.02.2022). A synthetic cumulative distribution of ED is calculated assuming a Gaussian error distribution (normalized to a total integral of 1) with standard deviation of ED. In the next step, this synthetic error frequency distribution is compared with the AODerror. We calculate and





subtract the mean bias from the AODerror distribution to make it more symmetric for direct comparison to the synthetic distribution (which by its definition is always symmetric). Bias correction results for S3A all, dual and singleN (0.07, 0.04 and 0.12, respectively) are shown in Figure 16.

Finally, we calculate an average correction factor for the synthetic distribution (and thus the prognostic uncertainties) in
relation to the mean-bias corrected error distributions as the ratio of the absolute means of both distributions. Corrections factors are different for all matchups, dual and singleN groups. A small correction is needed for all and singleN (0.80 and 1.1, respectively). For the dual group, the correction is stronger (0.67); ED should be lowered.

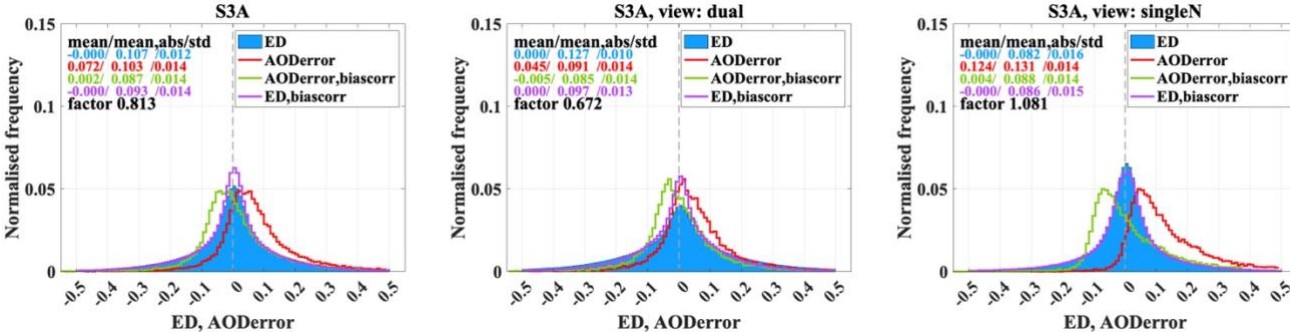

**Figure 16: Histograms of the ED (blue filled bars), AODerrors (red; with bias correction: green) and ED calculated from uncertainties (purple; scaled to best fit the mean-bias corrected error distribution) for all matchups (left panel), dual- (middle panel) and singleN (right panel) groups of matchups. Statistics, mean/mean,abs/std are mean over 'real ' values, mean over 'absolute' values and standard deviation, respectively, for histograms of the corresponding color.**

However, the correction method applied here is not equally improving ED in all ranges. The correction factor is biased by the number of pixels with small (<0.2) absAODerror. Thus, for those cases the correction works well; overestimated ED is lowered by 0.8/0.65 for all and dual groups. For absAODerror > ca.0.3, where ED is underestimated, correction degrades ED and increases disagreement between ED and AODerror. Possible solution can be in performing correction separately for different absAODerror ranges but setting specific relations for different groups between ED and absAODerror makes the analysis very
complicated.

### 6.2.4 Potential of the expected discrepancy

Sayer et al. (2020) suggested the analysis of the potential of the PU to discriminate between ('good' and 'bad') pixels with likely small / large errors. Instead of PU, we perform analysis of the ED, which, besides PU, includes uncertainties of the ground-based measurements.

To estimate the potential of ED, we plot the absolute errors below which 38% of all pixels are, as a function of binned ED (Figure 17). We then repeat this for the fractions 68% and 95%. These percentages relate to 0.5σ, 1σ, and 2σ (where σ is a





standard width) for normal error distributions in each bin (along the vertical axis). Theoretically expected values are shown as dashed lines in black, red, and blue. The number of pixels per ED bin is shown as a grey dashed line.

The percentile plots show a reasonable agreement (within statistical noise) with the theoretical lines of 38% and 68% for
majority of the validation points in the lower range of ED (up to 0.05-0.2) for all groups, with underestimation of the true error at higher values of ED for 38% and 68% lines. For the dual view case, ED overestimates the true error, while for the single view case the true error is higher than the ED prediction, especially at higher values of ED (ED>~ 0.2).

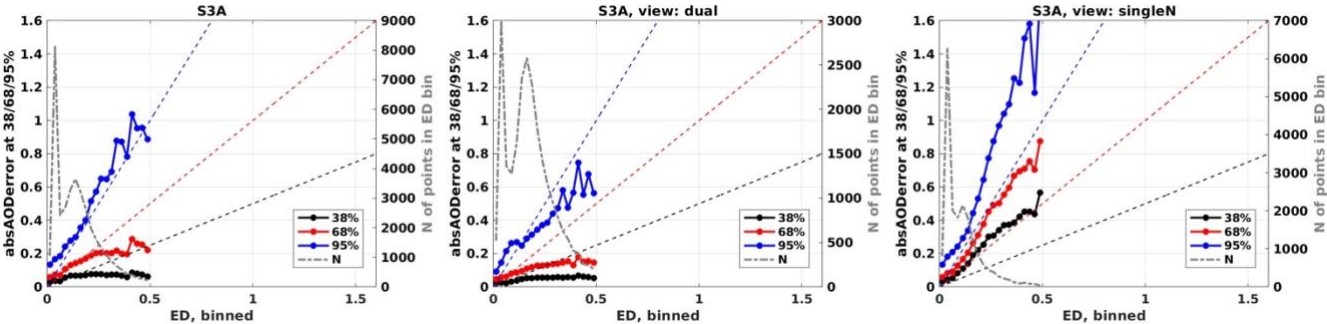

**Figure 17: Percentile plots of absAODerrors at 38% (black), 68% (red) and 95% (blue) as function of binned expected discrepancy.**

**6.3    Fine mode AOD and Fine Mode Fraction**

Fine mode AOD in the SY_2 product (syFMAOD) is provided at 550nm, while AERONET Fine mode AOD (aFMAOD) is provided at 500nm. As for $aAOD_{500}$, AOD spectral dependence (https://aeronet.gsfc.nasa.gov/new_web/man_data.html, O'Neill et al., 2003) was considered to convert $aFMAOD_{500}$ into $aFMAOD_{550}$.

Density scatter plots for the relation between syFMAOD and aFMAOD in the NH and SH, are shown in Figure 18 for S3A;
validation statistics are summarised in Table 5 for both S3A and S3B. The dispersion of points is higher in the NH. Validation results are considerably better in the SH: R is higher (0.67 vs 0.63 for the SH and NH, respectively), rms (0.15 vs 0.23) and bias (0.06 vs 0.14) are lower, slope (0.93 vs 0.70) is closer to 1. Analysis of the binned FMAOD shows that in both NH and SH, good agreement was observed between syFMAOD and aFMAOD for aFMAOD<1. At aFMAOD>1, syFMAOD is considerably underestimated in the NH. In the SH, only few aFMAOD values above 1 are measured. Validation statistics for
S3B are slightly better.





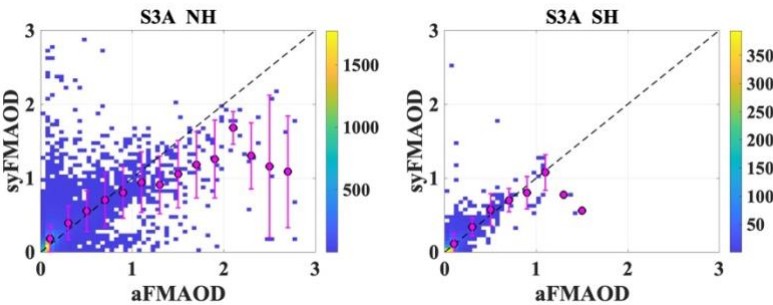

**Figure 18: Density scatter plots for S3A syFMAOD and corresponding aFMAOD for collocations available over the NH (left) and SH (right).**

Looking at the seasonal validation results, for both S3A and S3B, the correlation coefficient is slightly higher in MAM (0.65 /0.67, for S3A/S3B, respectively) and JJA (0.67/0.69) and lower (0.56/0.59) in DJF (Table 5; Figure S 9, Supplement). Bias is ca 0.1-0.12 and slightly higher (0.15/0.12) in JJA. The binned mean syFMAOD values are close to the 1:1 line for aFMAOD < 0.6-1 but fall below the line for higher aFMAOD.

**Table 5: For S3A and S3B, annual (for the globe, NH and SH) and seasonal (for the globe) validation statistics for syFMAOD.**

| Period | Region | N | | R | | rms | | std | | bias | | slope | |
|---|---|---|---|---|---|---|---|---|---|---|---|---|---|
| | | S3A | S3B | S3A | S3B | S3A | S3B | S3A | S3B | S3A | S3B | S3A | S3B |
| year | globe | 18145 | 18262 | 0,63 | 0,67 | 0,22 | 0,20 | 0,001 | 0,001 | 0,13 | 0,12 | 0,72 | 0,72 |
| | NH | 15883 | 15982 | 0,63 | 0,66 | 0,23 | 0,20 | 0,002 | 0,001 | 0,14 | 0,12 | 0,70 | 0,71 |
| | SH | 2262 | 2280 | 0,67 | 0,72 | 0,15 | 0,15 | 0,003 | 0,002 | 0,06 | 0,06 | 0,93 | 0,91 |
| DJF | globe | 2447 | 2418 | 0,56 | 0,58 | 0,21 | 0,18 | 0,004 | 0,003 | 0,12 | 0,10 | 0,59 | 0,53 |
| MAM | | 5832 | 5952 | 0,65 | 0,67 | 0,22 | 0,21 | 0,002 | 0,002 | 0,12 | 0,11 | 0,85 | 0,86 |
| JJA | | 7641 | 7579 | 0,67 | 0,69 | 0,23 | 0,20 | 0,002 | 0,002 | 0,1 | 0,13 | 0,71 | 0,70 |
| SON | | 2225 | 2313 | 0,49 | 0,66 | 0,22 | 0,16 | 0,004 | 0,003 | 0,12 | 0,10 | 0,56 | 0,62 |

Among selected regions (Figure 19), regional offset for all aerosol types is negligible (slightly positive) in Eur, Ind and NAW. In ChinaSE and AfN, an offset is increasing with increasing of aFMAOD over 0.5 and becomes more unstable (takes both positive and negative values).



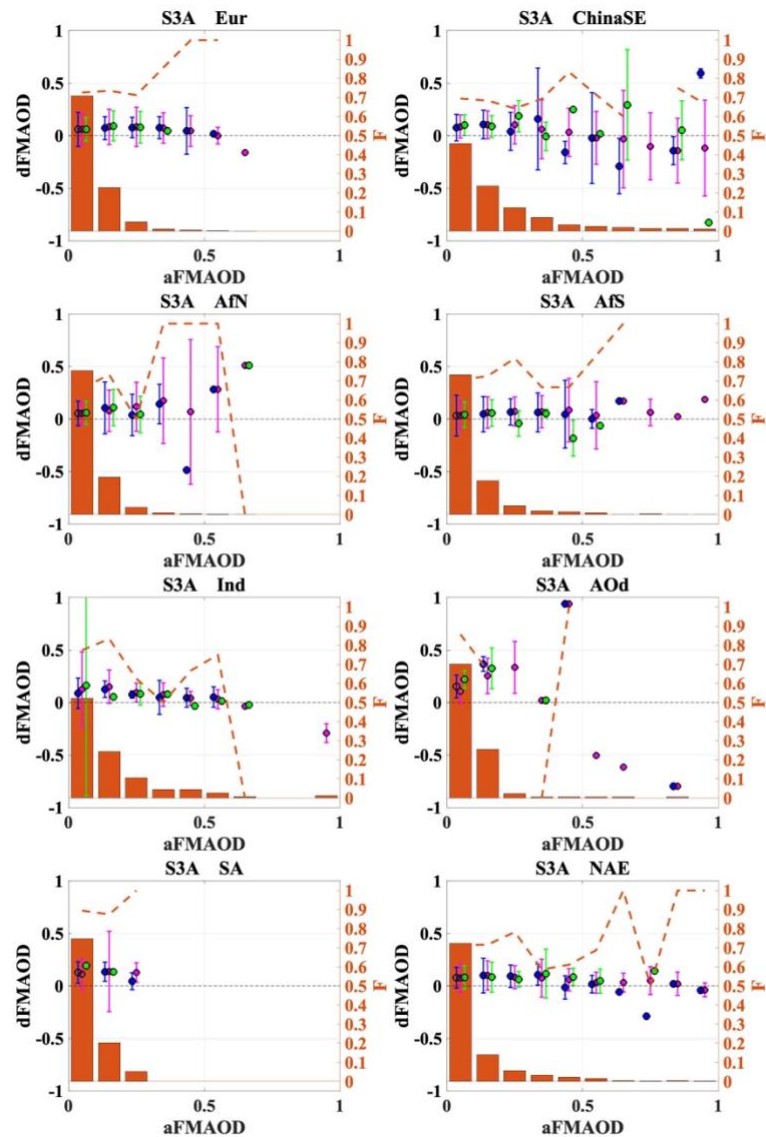


**Figure 19: Regional (for Eur, ChinaSE, AfN, AfS, Ind, AOd, SA, NAE) difference (dFMAOD) between syFMAOD and aFMAOD for selected aFMAOD bins: median bias (circles) and bias standard deviation (error bars) for all AOD types (purple), aerosol fine-dominated AOD (blue) and coarse-dominated AOD (green). The fraction (F) of points in each bin from the total number of matchups is represented by orange bars. The fraction of fine-dominated matchups in each bin is shown as orange dashed-line.**
**Results for other regions are in the Supplement (Figure S 10).**

SY_2 Fine Mode Fraction (syFMF), which is a fraction of syFMAOD from the total syAOD, was validated against AERONET Fine Mode Fraction (aFMF). Since syFMAOD is slightly overestimated, we expect that syFMF is overestimated as well. Density scatter plots for the relation between syFMF and aFMF in the NH and SH are shown in Figure 20 for S3A. In both

hemispheres, and thus globally, syFMF is overestimated in the aFMF range of 0-0.7; positive offset of 0.3-0.5 at low (<0.25)





aFMF is gradually decreasing. At aFMF>0.9, syFMF is slightly underestimated. Offset between syFMF and aFMF is slightly lower in the SH. For the NH/SH respectively, R is 0.34/0.42; bias is 0.56/0.49, slope is 0.28/0.37.

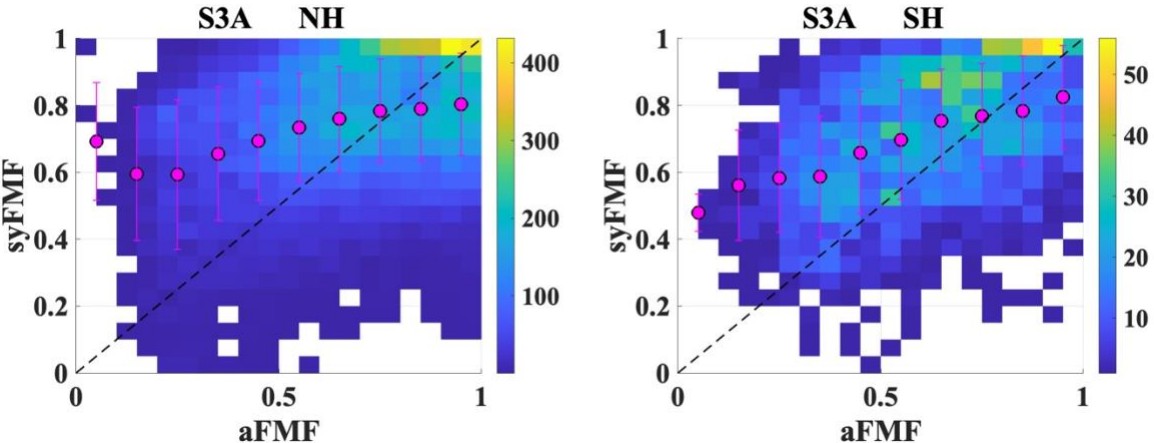

**Figure 20: Density scatter plots for S3A syFMF and corresponding aFM for collocations available over the NH (left) and SH (right).**

Scatter density plot between dFMF (which is defined as a difference between syFMF and aFMF) and aAOD is shown in Figure 21 for the NH and SH. In general, offset is higher at low AOD and decreases towards high AOD. The fraction of high (>0.05) overestimates is decreasing towards high AOD, while the fraction of high underestimates increases.

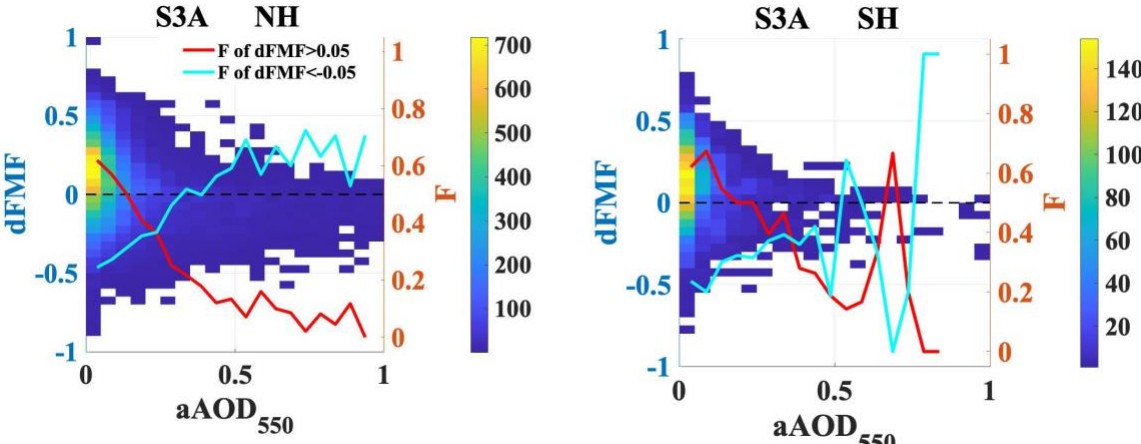

**Figure 21: Density scatter plot for the difference (dFMF) between syFMF and aFMF as a function of aAOD$_{550}$. Fractions of positive (dFMF>0.05, red line) and negative (dFMF<-0.05, blue line) overestimations per aAOD bin are shown.**

Regional dFMF (Figure 22) is positive (0.3-0.7) for low aFMF and decreasing gradually towards higher aFMF. At aFMF above 0.5-0.7, aFMF turns to negative (syFMF is underestimated). Similar tendency is observed for all chosen regions.



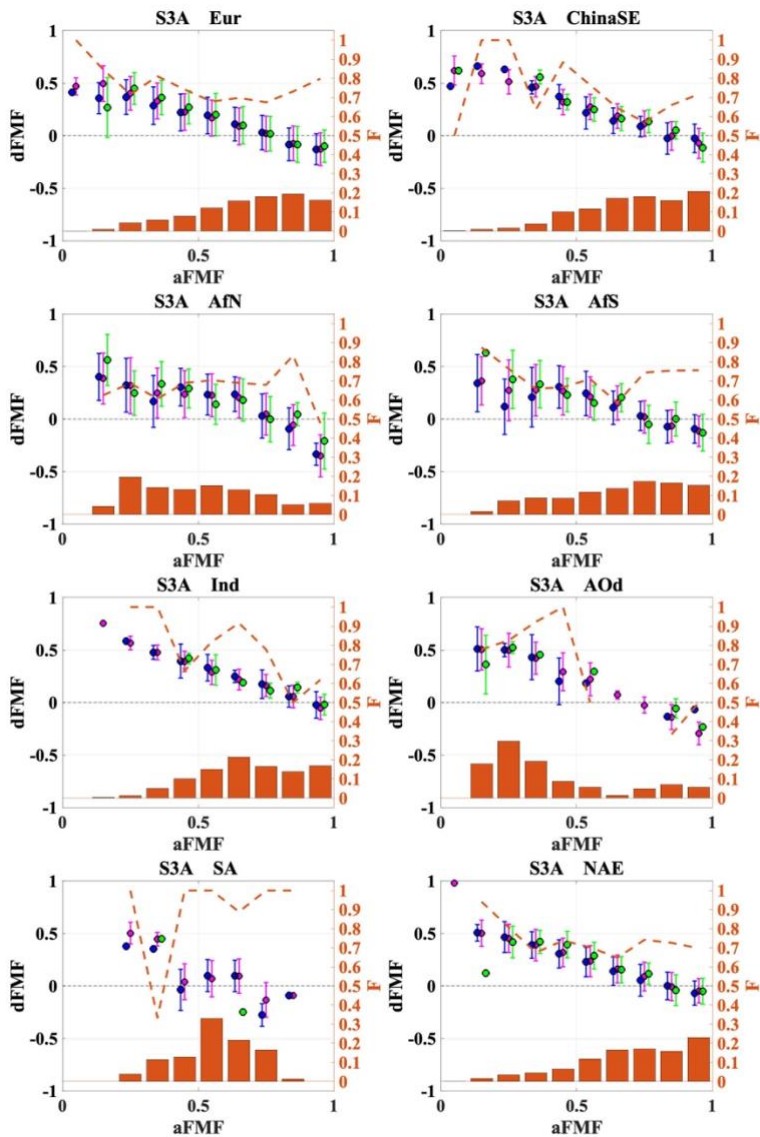

**Figure 22 Regional (for Eur, ChinaSE, AfN, AfS, Ind, AOd, SA, NAE) difference (dFMF) between syFMF and aFMF for selected aFMF bins: median bias (circles) and bias standard deviation (error bars) for all AOD types (purple), aerosol fine-dominated AOD (blue) and coarse-dominated AOD (green). The fraction (F) of points in each bin from the total number of matchups is represented by orange bars. The fraction of fine-dominated matchups in each bin is shown as orange dashed-line. Results for other regions are in the Supplement (Figure S 11).**

## 6.4    Ångström exponent

The Ångström exponent, AE, is often used as a qualitative indicator of aerosol particle size. Synergy AE (syAE) is calculated in the spectral interval 550-865 nm, while AERONET AE (aAE) is provided for 500-870 nm. The difference between $AE_{550-865}$ and $AE_{500-870}$ depends on the aerosol type and may be as high as 5-10% of AE (personal estimations). This difference must be considered for the interpretation of the evaluation results.





Scatter plots between $syAE_{550-865}$ and $aAE_{500-870}$ for S3A for all matchups and different groups of matchups are shown in Figure 23 and corresponding validation statistics are shown in Table 6. Two "clouds" of satellite/AERONET AE matchups are clearly observed. The first cloud is in the aAE interval of 1-1.6 and syAE around 1.2. In that interval, the cloud of pixels is located around the 1:1 line, which means that the agreement between syAE and aAE is quite good. The second "cloud" is in the aAE interval [1.4 1.9] and syAE around 2. In that interval, syAE is overestimated by 0.3-0.6.

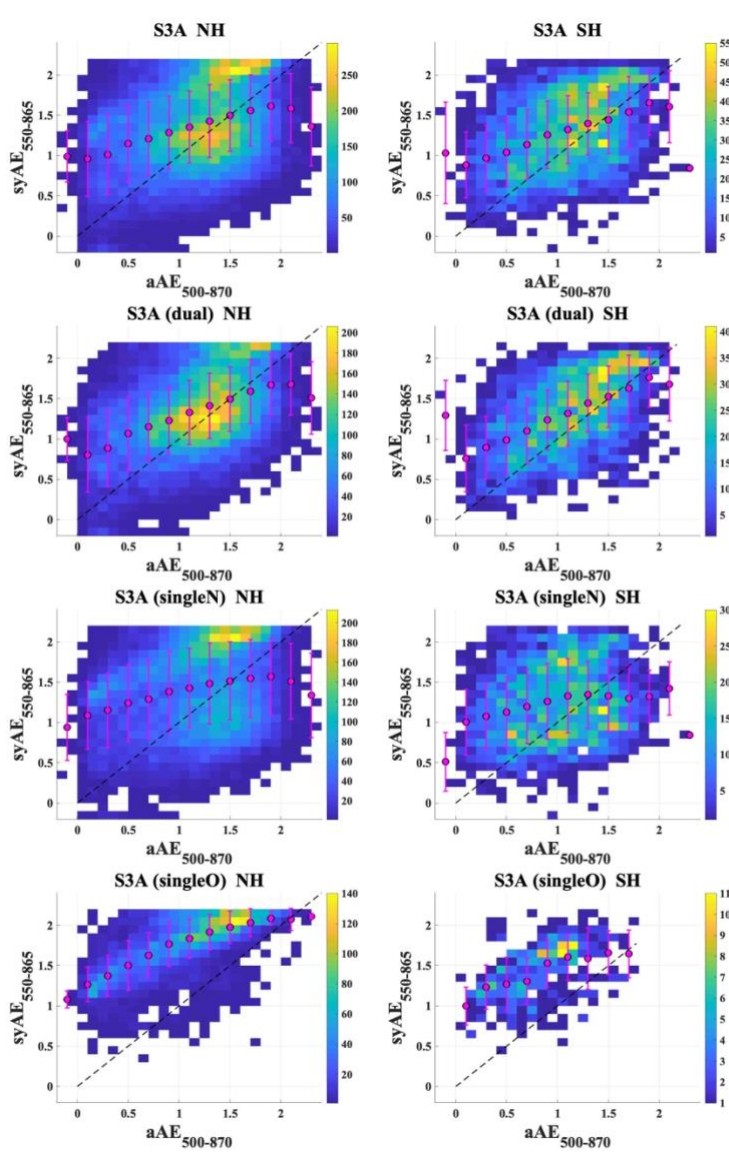


**Figure 23: Scatter plots between $syAE_{550-865}$ and $aAE_{500-870}$ for S3A for the NH and SH (panels left and right, respectively) for different groups of products (top-down: all, dual, singleN and singleO).**





For the whole global product, correlation coefficients between $syAE_{550-865}$ and $aAE_{500-870}$ are quite low, 0.35/0.34, rms is high, 0.57/0.58 for S3A/S3B, respectively. Validation statistics are slightly better for the dual product. The singleO product shows

better correlation, but worse rms and std. Validation statistics are better in the NH for all matchups and the dual product. For the single view groups (singleN and singleO), no difference in validation results was revealed between the NH and SH.

**Table 6: Validation statistics (number of points, N; correlation coefficient, R; root mean square, rms; standard deviation, σ; bias and slope defined with linear regression applied to all available matchups) for S3A and S3B $syAE_{550-865}$ for the globe, NH and SH for the whole period for all matchups and for three groups of matchups, defined with the processor applied.**

| Group | Area | N | | R | | rms | | std | | bias | | slope | |
|---|---|---|---|---|---|---|---|---|---|---|---|---|---|
| | | S3A | S3B | S3A | S3B | S3A | S3B | S3A | S3B | S3A | S3B | S3A | S3B |
| all | globe | 38469 | 38920 | 0,35 | 0,34 | 0,57 | 0,58 | 0,003 | 0,003 | 0,94 | 0,97 | 0,36 | 0,35 |
| | NH | 32939 | 33323 | 0,34 | 0,33 | 0,58 | 0,58 | 0,003 | 0,003 | 0,96 | 1,00 | 0,35 | 0,33 |
| | SH | 5530 | 5597 | 0,39 | 0,42 | 0,54 | 0,53 | 0,007 | 0,006 | 0,86 | 0,83 | 0,41 | 0,44 |
| dual | globe | 25161 | 25862 | 0,46 | 0,44 | 0,51 | 0,51 | 0,003 | 0,003 | 0,81 | 0,83 | 0,46 | 0,43 |
| | NH | 21487 | 22050 | 0,45 | 0,42 | 0,51 | 0,52 | 0,003 | 0,003 | 0,82 | 0,86 | 0,45 | 0,41 |
| | SH | 3674 | 3812 | 0,51 | 0,53 | 0,49 | 0,48 | 0,007 | 0,007 | 0,73 | 0,68 | 0,53 | 0,57 |
| singleN | globe | 20039 | 19972 | 0,25 | 0,25 | 0,64 | 0,66 | 0,004 | 0,004 | 1,10 | 1,15 | 0,26 | 0,25 |
| | NH | 17161 | 17115 | 0,24 | 0,23 | 0,65 | 0,67 | 0,005 | 0,005 | 1,12 | 1,18 | 0,26 | 0,24 |
| | SH | 2878 | 2857 | 0,19 | 0,24 | 0,61 | 0,61 | 0,010 | 0,010 | 1,06 | 1,05 | 0,20 | 0,25 |
| singleO | globe | 5246 | 5402 | 0,68 | 0,67 | 0,79 | 0,80 | 0,005 | 0,005 | 1,25 | 1,27 | 0,48 | 0,46 |
| | NH | 4906 | 5033 | 0,69 | 0,67 | 0,79 | 0,80 | 0,005 | 0,005 | 1,28 | 1,31 | 0,47 | 0,45 |
| | SH | 340 | 369 | 0,48 | 0,52 | 0,70 | 0,69 | 0,020 | 0,018 | 1,06 | 1,05 | 0,45 | 0,46 |


Regional analysis (Figure 24) reveals considerable differences in syAE evaluation results for regions with different surface type and aerosol properties. Footprints for the frequency of matchups at certain AE ranges (density value on the scatter plot) follow the "cloudy" shape in regional scatter density plots. Location of the "clouds" along x-axis (aAE) is specified by prevailing aerosol types in those regions. The "cloudy" shape of the footprint often ruins validation statistics (Table 7), which

should be interpreted with consideration of the matchup's footprint Figure 24.

**Figure 24: Regional scatter density plots between syAE$_{550-865}$ and aAE$_{500-870}$. Regions are defined in Figure 5.**

Over land, syAE is often overestimated in aAE range of 1.3-1.7, except for AsW, where the fraction of "good" (close to the 1:1 line) pixels is as high as fraction of overestimated syAOD. In AfN, low AE, which is typical for that region characterized by a high fraction of dust particles, is often highly overestimated. Location of the smaller "cloud" of pixels near the 1:1 line proves a good quality of syAE. Dense cloud of "good" matchups is located near the 1:1 line in NAW. However, R (Table 7) is low in that region, because, as mentioned above, the shape of the "good" pixels has a shape of a cloud and statistics are defined by outliers which are distributed evenly in all directions from the "cloud". In oceanic regions with possible transport of dust aerosols, syAE is often underestimated. The low number of matchups in AOb region (N = 22) doesn't allow making a solid conclusion on the syAE quality in this region.





Regional validation statistics for syAE (number of collocations per area N, R, rms, bias, slope) are summarised in Table 7. Rms is above 0.5 for all areas, except AsW. syAE bias is 0.35 in AsW, 0.7-1.0 on Eur, Bor, AsN, Aus, Afn, AfS and above 1.0 in the other regions. Slope is 0.72 in AsW, 0.4-0.5 in Bor, AsN, AfN, AfS and below 0.4 in the other regions.

Combination of the results shown in Figure 24 and summarised in Table 7, prove, in general, good quality of syAE, though R is often low. Positive syAE bias is clearly seen in AfN region for all aAE ranges and in the neighbouring AOd region for low aAE.

**Table 7: For S3A, syAE$_{550-865}$ validation statistics: number of points, N; correlation coefficient, R; root mean square, rms; standard deviation, σ; bias and slope defined with linear regression applied to all available matchups.**

| Region | N | R | rms | sigma | bias | slope |
|---|---|---|---|---|---|---|
| Eur | 9225 | 0,28 | 0,55 | 0,006 | 0,99 | 0,29 |
| Bor | 1737 | 0,35 | 0,58 | 0,014 | 0,80 | 0,43 |
| AsN | 785 | 0,39 | 0,52 | 0,017 | 0,88 | 0,44 |
| AsE | 2392 | 0,17 | 0,6 | 0,010 | 1,35 | 0,18 |
| ChinaSE | 2322 | 0,23 | 0,59 | 0,010 | 1,26 | 0,24 |
| AsW | 1789 | 0,57 | 0,45 | 0,011 | 0,35 | 0,72 |
| Aus | 629 | 0,29 | 0,54 | 0,019 | 0,84 | 0,29 |
| AfN | 3043 | 0,44 | 0,64 | 0,010 | 0,71 | 0,52 |
| AfS | 3302 | 0,42 | 0,54 | 0,009 | 0,79 | 0,43 |
| SA | 136 | 0,00 | 0,56 | 0,048 | 1,15 | 0,00 |
| NAE | 6156 | 0,41 | 0,61 | 0,006 | 1,12 | 0,38 |
| NAW | 5850 | 0,17 | 0,52 | 0,007 | 1,10 | 0,13 |
| Ind | 605 | 0,17 | 0,61 | 0,017 | 1,42 | 0,17 |
| Aod | 369 | 0,11 | 0,81 | 0,032 | 1,06 | 0,08 |
| Aob | 22 | 0,54 | 0,61 | 0,056 | 1,01 | 0,43 |

**7    Validation over ocean**

Being performed on-board ships, MAN AOD measurements are irregular. S3A and S3B collocations with MAN for the period 01.2020-09.2021 are shown in Figure 25. Altogether, 105 matchups have been found for S3A and 95 matchups for S3B. Note, that about half of the collocations are observed near coastal zones. Since the number of validation points is low, we show in Figure 26 scatter plots and validation statistics for both S3A and S3B.

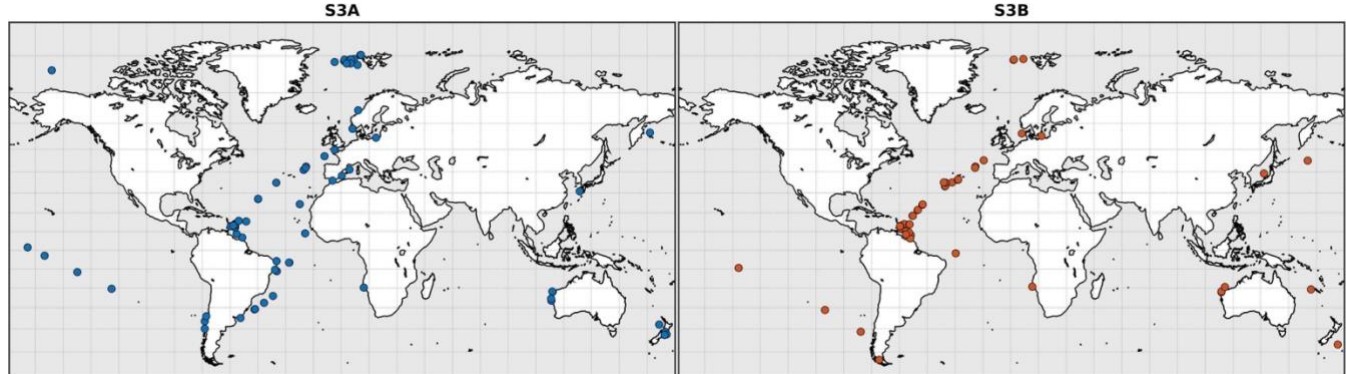


**Figure 25: Collocations of S3A (left) and S3B (right) with MAN, 01.2020-09.2021**

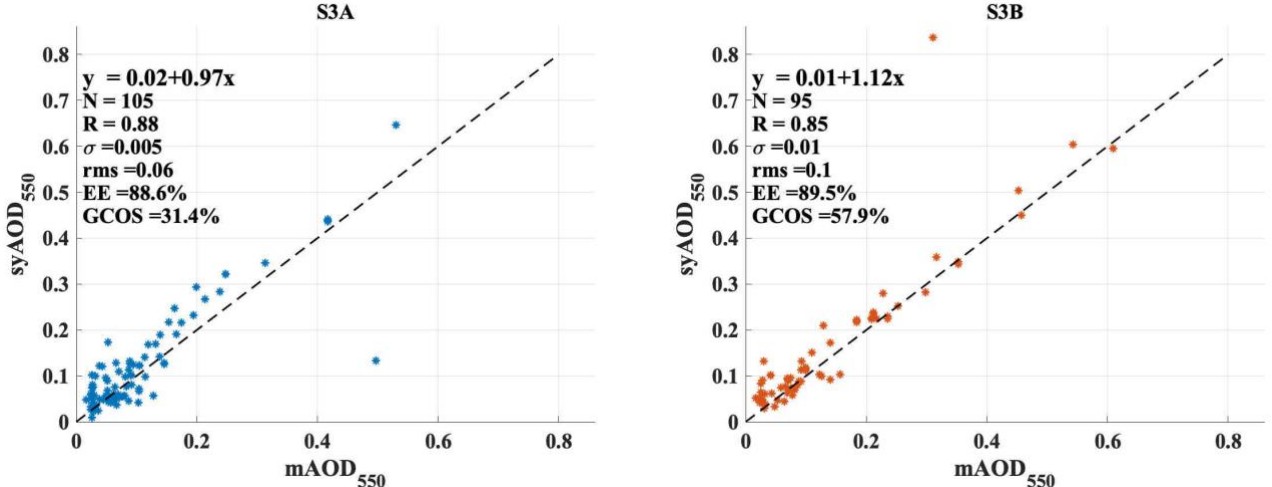

**Figure 26: Scatter plots between S3A and S3B syAOD and MAN AOD (mAOD) with validation statistics.**

Results for both instruments confirm a good performance of the retrieval algorithm over ocean. For S3A/S3B, correlation
coefficients are 0.88/0.85, fractions of pixels in the EE are 88.6/89.5 %. An offset with MAN AOD (mAOD) is slightly higher
for S3A (0.02/0.01), while rms is slightly higher for S3B (0.06/0.1).

One value from each product, S3A and S3B, can be considered as a clear outlier: S3A over the Baltic is underestimated, S3B
over the Caribbean Sea is overestimated. The removal of these outliers from the validation exercise improves validation
statistics: correlation increases to 0.95/0.97, rms decreases to 0.04/0.03, fractions of pixels in the EE increases to 89.4/92,4 %
for S3A/S3B, respectively.





## 8    SY_2 AOD spatial performance relative to MODIS Terra DT&DB AOD product

### 8.1    Methods

The coverage of ground-based reference data is limited. To better evaluate a spatial distribution of the satellite retrieved AOD, the inter-comparison with other satellite products is necessary. The satellite product chosen as a "reference" must fulfil several
criteria, e.g.:

(i) overpass time as close as possible to Sentinel-3 to avoid possible different aerosol and cloud conditions;

(ii) wider swath (for the reference product), which allows considering most of the pixels from the tested product in the analysis;

(iii) similar resolution, which allows pixel-to-pixel intercomparison.

Considering these criteria, the MODIS Terra DT&DB AOD product has been chosen as a reference for evaluation of the SY_2
$AOD_{550}$ product.

MODIS Terra DT&DB AOD product fulfils two out of three criteria mentioned above:

(i) The Sentinel-3 orbit is a near-polar sun-synchronous orbit with a descending node equatorial crossing at 10:00 am Mean Local Solar time. The MODIS Terra satellite is crossing the equator on descending passes at 10:30 -10:45 AM.

(ii) SLSTR dual view swath centred on the sub-satellite track is 740 km wide, with a single view swath width of 1470 km.
OLCI instrument covers a swath width of 1,270 km. MODIS Terra has a viewing swath width of 2,330 km.

The (iii) criteria is not fulfilled since MODIS and SY AOD products are provided at different resolutions. The resolution of the SY_2 product is 4.5x4.5 $km^2$, while the MODIS AOD daily product is available at 3km, 10km and 1° resolution, MODIS monthly product is available at 1° resolution. Thus, to fulfil the third criterion, we re-grided daily SY_2_AOD product to 1° resolution for an area of interest (AOI) and calculated monthly aggregates. One degree grid resolution was chosen to mitigate
collocation uncertainties, smooth the data and minimise the processing time.

The inter-comparison of satellite monthly AOD is considered from two points of view. For algorithm performance inter-comparison, only the spatio-temporally collocated pixels from the two products were considered (used in monthly aggregates). For climate studies, when monthly products are utilized (for, e.g., model evaluation, trend analysis), inter-comparison should be performed for the products built on all points available for each instrument, respectively.
SY_2 and MODIS Terra AOD products were inter-compared over the area shown in Figure 27. To inter-compare AOD products (and thus algorithm performance) in different environments (e.g., surface type, aerosol type, aerosol loading), sub-regions shown in Figure 28 (top right) were chosen (see Table S2 for details).

### 8.2    Inter-comparison of daily AOD products

All pixels available in S3A SY_2_AOD and MODIS Terra L3 daily AOD550 products, collocated products and differences
between collocated products are shown for a selected area of interest (AOI) for 26th of February 2020 (Figure 27). Because of the wider swath, MODIS has larger coverage than S3A. Thus, when collocating two products for closer intercomparison, more pixels from the MODIS product are removed.





For the products containing all original pixels for each instrument respectively, the SY_2 AOD mean over the AOI is higher than MODIS Terra AOD (0.35/0.21 for S3A/ MODIS, respectively). Mean AOD over land and over ocean are also higher for

S3A. For collocated products, mean (over the AOI) AOD for S3A and MODIS, as well as AOD over ocean come very close to each other. Over land, mean AOD is slightly lower for S3A for collocated pixels.

For the chosen day, for S3A, a sharp transition between AOD retrieved over land and ocean at the west coast of Africa is revealed. This feature is clearly seen in the S3A and MODIS AOD difference plot. The large AOD gradient in S3A data is observed over Nigeria; the inconsistency with MODIS data reaches above ±0.5 AOD in this area.


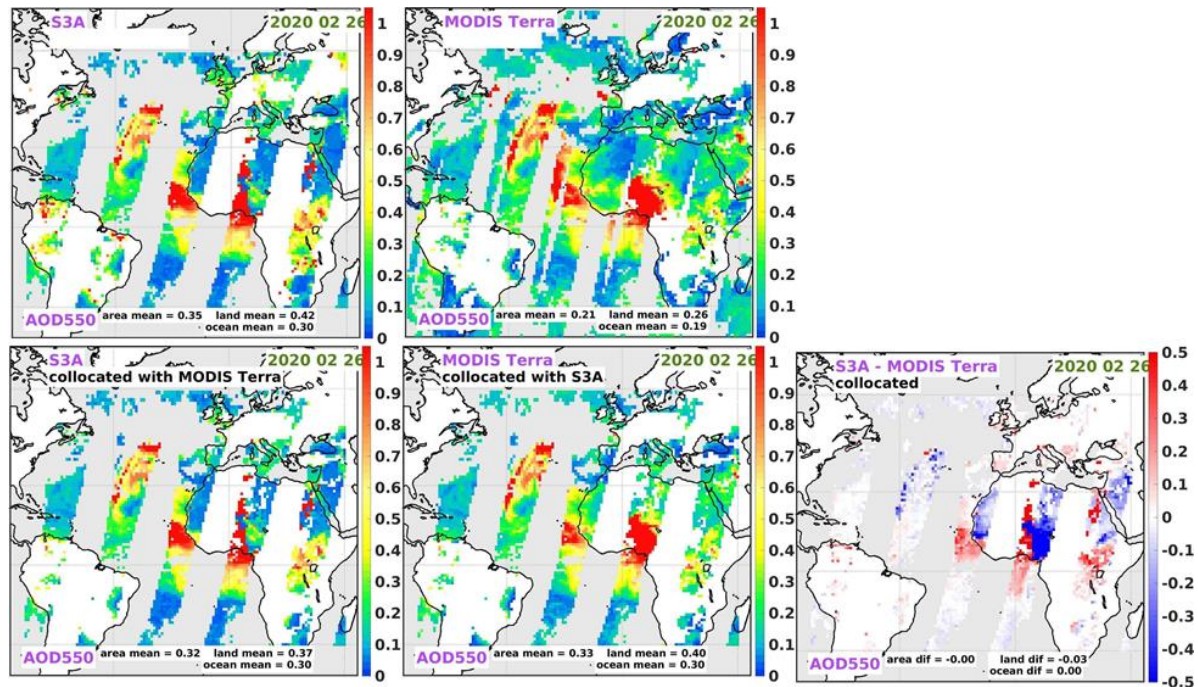

**Figure 27: For 2020-02-26, Upper panel: All pixels available in products, S3A SY_2_AOD (left), MODIS Terra AOD (middle). Lower panel: Pixels existing in both products (collocated products), S3A SY_2_AOD (left), MODIS Terra AOD (middle) and difference between S3A SY_2_AOD and MODIS Terra AOD (right). For each sub-plot, statistics for AOD$_{550}$ (mean AOD for the whole area**
**and separately for land and ocean) are shown.**

For the whole year 2020, S3A SY_2 and MODIS AOD$_{550}$ pixel-level inter-comparisons of 1°x 1° daily products for chosen sub-regions are shown as density scatter plots in Figure 28.

In Europe region, which includes parts of Eastern and Southern Europe and Middle East, AOD is low (<0.4) in both products, in general. However, several outliers are observed in SY_2 product (SY_2 AOD is in the range 1-4, while MODIS AOD is

below 0.5). A possible reason for disagreement can be that SY_2 AOD was retrieved in cloud edge, while MODIS has been retrieving AOD in clear sky condition (given ca 30 min difference between overpasses). If this is true, SY_2 cloud screening should be improved to better distinguish between aerosol and clouds in cloud edge areas. The outlier cases should be studied separately to better understand a reason for disagreement.



In the desert area the disagreement between the two products is most significant. For MODIS AOD in the range 0-0.8 most of
the SY_2 pixels have AOD<0.2, while there are also a considerable number of SY_2 pixels with AOD in the range 1-4. For
MODIS AOD above 0.8, SY_2 AOD is often low, which is confirmed with averaged over MODIS AOD bins results (magenta
dots in Figure 28). The high surface reflectance typical to this area is challenging for aerosol retrieval. The large variance
observed in the AOD comparison indicates that a more detailed inter-comparison including the surface reflectance values
retrieved by each algorithm should be performed. Over clean ocean and ocean+dust sub-regions, an agreement between SY_2
and MODIS AOD is quite good, though SY_2 AOD is slightly lower when MODIS AOD is above 0.8 and above 1.2,
respectively.

In coast+dust area (over which biomass burning aerosols can be transported occasionally), AOD averaged over bins are biased
slightly positive for AOD<1.2, which results from SY_2 positive outliers, while for AOD>1.2 SY_2 AOD is often much lower
than MODIS AOD, thus binned averaged AOD is biased negative.

The footprints for SY_2 and MODIS AOD look similar in the two areas with seasonal contribution of biomass burning aerosols
(Africa,BB and S.America,BB ). An agreement between SY_2 and MODIS is good for MODIS AOD below 1.2. Above that
threshold, SY_2 AOD is on average lower.

Overall, the majority of data is in the low AOD range, where agreement is decent (with SY_2 slightly high biased), but at
higher AOD there is much more variance (partly due to the scarcity of data), and in general a slight low-bias for SY_2.

Seasonal comparison is shown in Figure S 12, supplement. Annual and seasonal statistics for SY_2 and MODIS Terra for all
daily pixel AOD inter-comparison are summarised in the supplement (Table S2, Table S3).





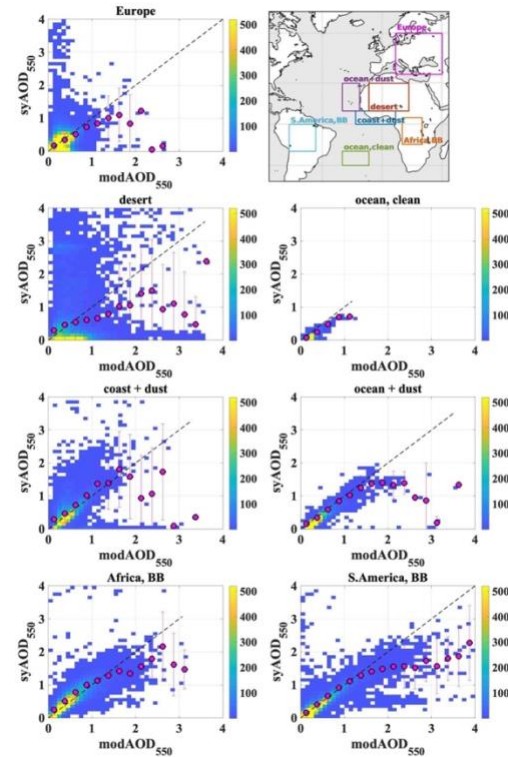

**Figure 28: Density scatter plots for MODIS Terra and S3A SY_2_AOD L3 daily collocated products for 2020 for the sub-regions shown in the top right corner. Statistics are summarised in the Supplement (Table S1, Table S2).**

**8.3    Spatial inter-comparison of monthly, seasonal and annual S3A and MODIS Terra AOD products**

Two types of monthly datasets have been created from SY_2_AOD and MODIS Terra daily data to study the differences at monthly/seasonal/annual (MSA) level.

In the first monthly dataset, all pixels available in SY_2_AOD and MODIS Terra daily products have been used to build a monthly aggregate, respectively for each instrument. Inter-comparison of these 'all pixels' monthly aggregates (which are

similar to the official monthly products provided for users) is important because it will help in, e.g., understanding the difference in climate data records which are built from the provided monthly AOD products which include all available data.

A second monthly dataset, 'collocated' product, has been aggregated using only collocated daily pixels. Inter-comparison of 'collocated' monthly aggregates shows the difference in monthly AOD based on differences in instruments and approaches.

Annual AOD from 'all pixels' and 'collocated' monthly datasets for SY_2_AOD and MODIS Terra, respectively, and the

corresponding differences are shown in Figure 29. Seasonal plots for collocated aggregates and difference between them are shown in Figure 30. Statistics for difference plots (area/land/ocean means) have been calculated from pixel-to-pixel difference, but not as difference between the AOD averaged over AOI, land and ocean.

Differences between SY_2_AOD and MODIS Terra MSA AOD exist in both 'all pixels' and 'collocated' datasets (annual: Figure 29; seasonal: Figure 30 and Figure S6; monthly results are not shown). For both datasets, SY_2 AOD averaged over



AOI is higher for the whole area, as well as for land and ocean. The difference is smoother for 'all pixels' datasets. Even though difference plots show that regional offset between two datasets is often within GCOS requirements of AOD quality (0.03) over ocean (SY_2 AOD is in general lower) and whole AOI, difference in AOD over land is often higher (up to 0.11 as averaged over AOI in DJF, 'all pixels' dataset).

Regional differences in seasonal AOD from the 'collocated' dataset are considerably higher (Figure 30). For all land sub-

regions (except for 'desert', JJA), S3A AOD is higher than MODIS AOD. The offset is highest for 'coast+dust' region in DJF and for 'Africa,BB' region in SON (0.18 and 0.15, respectively). General tendency of decreasing offset towards JJA months has been observed. However, though the offset is often high, time series for both products are within an overlap (grey area) of the standard deviations for individual products. Highest negative offset (between 0.05 and 0.1) is observed in JJA in the 'desert' region. Regional differences in seasonal AOD from the 'all pixels' dataset are less scattered (Figure S6, Supplement).

For the open ocean regions ('ocean, clean' and 'ocean+dust'), S3A AOD is in general lower than MODIS AOD for all MSA; the exceptions are January and February in 'ocean+dust' region (Figures not shown). In the annual scale, the offset between S3A and MODIS AOD is -0.02 for 'ocean+dust' and -0.03 for 'ocean, clean'. AOD in 'collocated' dataset is higher compared to 'all pixels' dataset for both S3A SY_2 and MODIS Terra. Comparing with 'all pixels', 'collocated' SY_2 AOD product looks less smooth over Northern Africa in DJF and MAM.

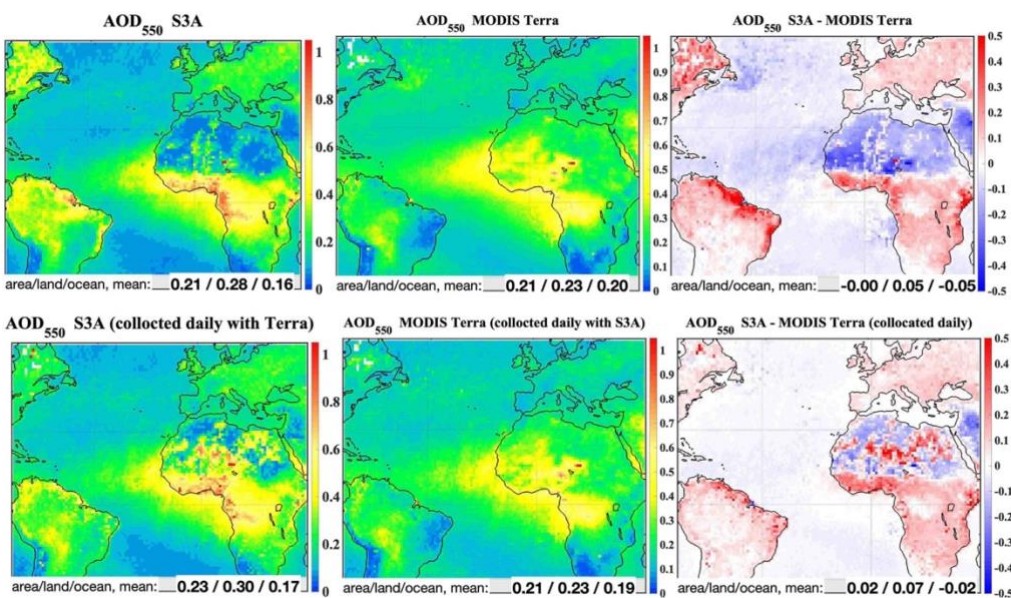


**Figure 29: For year 2020, annual S3A SY_2_AOD (left panel), MODIS Terra (middle panel) AOD and difference in between S3A and MODIS Terra (right panel) AOD. Annual means are calculated from monthly aggregates combined from all data available in each product (upper panel) and pixels of collocated daily AOD (lower panel). AOD mean and difference between SY_2 and MODIS**
**AOD for the whole area, as well as separately for land and ocean, are shown on the maps.**

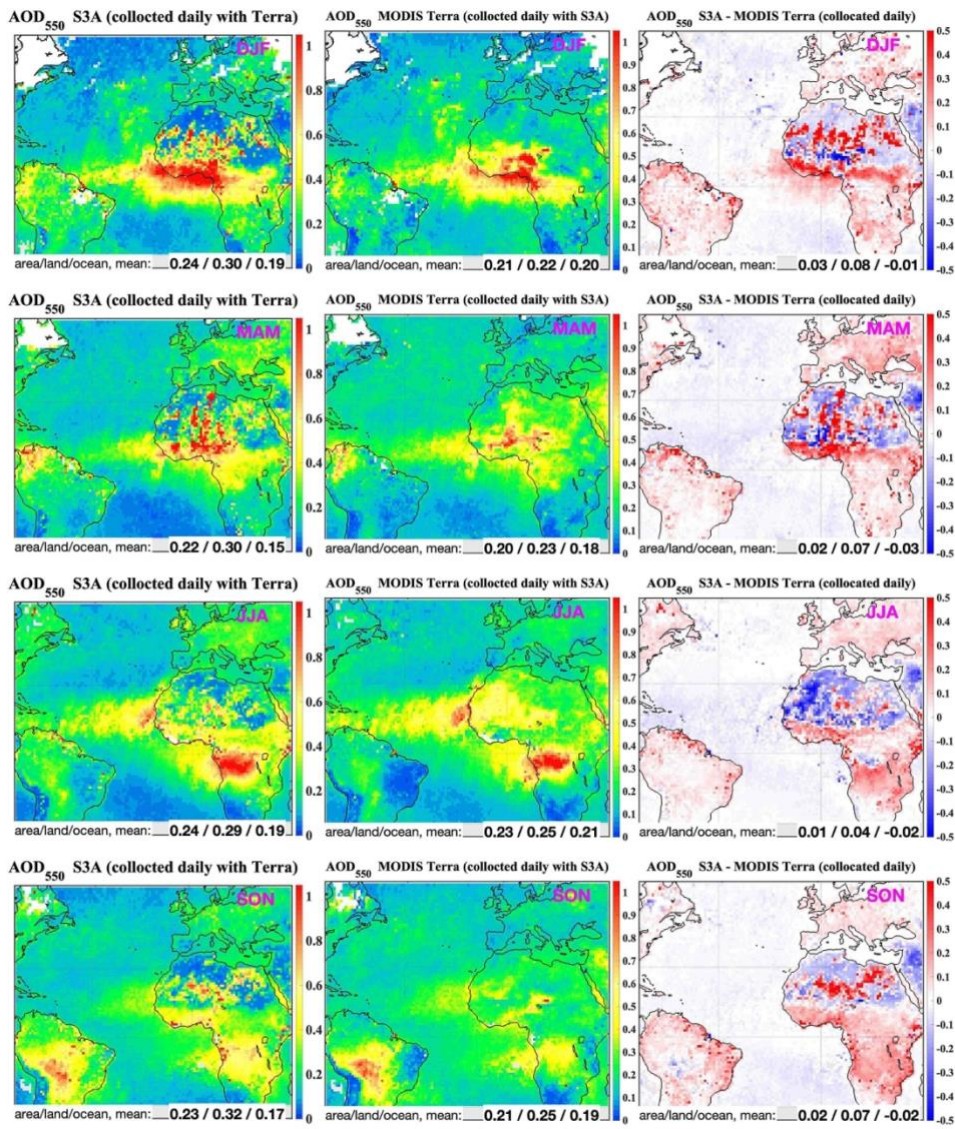

**Figure 30: Seasonal (top down: DJF, MAM, JJA, SON) S3A (left panel), MODIS Terra (middle panel) AOD$_{550}$ and difference in AOD$_{550}$ between S3A and MODIS Terra (right panel). From monthly aggregates created from collocated daily S3A and MODIS Terra AOD products.**

## 9   Conclusions and recommendations for future evolution

We have presented the first validation of a new SYNERGY global aerosol product, derived from the data from the OLCI and SLSTR sensors onboard the Sentinel-3A and -3B satellites. Combined, the two satellites provide close to daily global coverage and provide aerosol measurements with a latency of 2-3 days. In this study we have compared the aerosol product with ground-based photometer data from four networks: AERONET, SKYNET, SURFRAD, and MAN, and with MODIS combined Dark



Target and Deep Blue algorithms. The aim of this study was to provide global characterisation of the current aerosol retrieval, and to guide future algorithm development.

Over ocean, the performance of SYNERGY retrieved AOD is good and consistent with reference MAN dataset (rms ~0.05), although the MAN validation has a limited set of higher AOD examples. Against MODIS, agreement is good, although SYNERGY AOD shows lower values at high AOD (>1.5) in dust regions, potentially indicating cloud screening improvement

needed to correctly detect high dust levels.

Over land, overall performance has a much higher rms error, approximately 0.25 when compared to AERONET. Overall AERONET correlation is ~0.6. Reduced performance over land is expected since the surface reflectance and angular distribution of scattering are higher, and they are more difficult to treat over land than over ocean. However, the results show that these statistics are affected by a large number of outliers. Inspection of these outliers and patterns of disagreement with

MODIS indicate possible reasons and targets for future algorithm evolution. The main causes are (i) poor screening of snow/ice covered surfaces, (ii) inadequate cloud screening in some regions. For example, in tropical forest areas, care needs to be taken to fully exclude any pixels containing clouds, including sub-pixel clouds in either nadir or oblique view. In addition, removal of cloud edge pixels (cloud free pixels next to cloud masked pixels) should be considered. Bright desert surfaces also have less stable retrieval, with land/ocean contrast suggesting high values in dust plumes are underestimated over land. It is clear that

retrievals using dual view give higher quality, making use of more information to allow less reliance on surface spectral assumptions.

The retrieval of Angstrom exponent, related to aerosol size distribution, shows spatial correlation with expected sources but generally overestimates AE for cases where AERONET Angstrom is low, resulting in overall high bias. This is dependent on the retrieval of fine mode fraction in the algorithm, which needs to be investigated further and improved. Evaluation of the

per-retrieval uncertainty indicated good correlation with measured error distributions, with overprediction of expected error in dual view case, and underprediction in single view case. Evaluation of the uncertainty propagation is difficult in the presence of outliers which do not fit the algorithm assumptions, where we see a tail of higher errors, for example related to undetected cloud in the input data.

**Author contribution**

CH, LS and SD created the original research framework and provided research direction. MD, CH established a data base. LS developed a validation strategy, wrote the software and performed the analysis. PK co-wrote the software. LS, TV, PN, 880 CH, SS and SD co-wrote the manuscript.

**Competing interests**

The contact author has declared that neither they nor their co-author has any competing interests.

**Funding**

This work has been performed in the frame of the LAW project under the European Space Agency Contract 4000129877/20/I-BG.





**Data access**

SY_2_AOD product: https://scihub.copernicus.eu/dhus/#/home , last access 13.03.2022

SY_2_AOD product validation matchups: https://law.acri-st.fr/home, last access 10.01.2022

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
