# Peer review of "Extended validation and evaluation of the OLCI-SLSTR Synergy aerosol product (SY\_2\_AOD) on Sentinel-3"

_Atmospheric Measurement Techniques, 2022_

## Referee Comment (RC2)

**Review of "Extended validation and evaluation of the OLCI-SLSTR Synergy aerosol product (SY_2_AOD) on Sentinel-3" by Sogacheva et al.**

*Summary:*

This paper presents the synergy AOD product from Sentinel-3 and its evaluation against a set of other global AOD products. This is obviously product of a thorough comparison, from the use of validations against AERONET, MAN (and SURFRAD and SKYNET in supplement), and MODIS datasets, and the breadth and level of detail of the manuscript shows it. This is a high-quality manuscript and should be published in AMT, and will likely be used as reference for many other validation of satellite aerosol products. While this manuscript is long, it is obviously needed, and the quality of the work is appreciated.

I recommend this paper to be published, but after addressing these issues:

- The linear fitting scheme is not well identified, or may not be appropriate for AOD fitting, and by the manuscript's own analysis (section 6.1.5), this matters for quantifying the overall fit. See the general comment #6. This would not be brought up as major concern except for the fact that it is highlighted in the manuscript already.
- There are numerous errors in formatting throughout the manuscript which detracts from the quality.
- The description of the retrieval methodology is unclear. How does the retrieval of AOD at multiple wavelength and single scattering albedo is achieved through fitting of AOD at only wavelength (550 nm)?

*General Comments:*

1. Several language issues are found within the abstract, and there is need for more quantitative indication in the abstract instead of the subjective descriptions (see specific comments below)
2. Throughout the document the date format does not seem to meet the AMT standard of "Date and time: 25 July 2007 (dd month yyyy), 15:17:02 (hh:mm:ss)", particularly evident in the paragraph at line 79-89. See the guidelines: https://www.atmospheric-measurement-techniques.net/submission.html#math
3. How much time is passed between measurements in the oblique and nadir view? And how does that impact the aerosol retrieval, particularly near clouds?
4. The retrieval dictates the retrieval of AOD and its fine mode at 550 nm, however returns many more parameters, including single scattering albedo, at various wavelengths. This is poorly described, and is both referred to as 'aerosol properties retrieved' and 'intended as diagnostics' (section 2.2.2). Please clarify what these properties are, and how they are retrieved, especially when only fitting to AOD and fine mode AOD at 550 nm.
5. Many references and citations are only links to websites, many of which should be replaced by the appropriate citation, and many are missing the date accessed.
6. The type of linear regression is not identified, and this matters for AOD comparisons. Reference to a 'linear regression' between the aAOD and syAOD is presented, however it seems to imply

the use of the Ordinary-Least-Squares (OLS) commonly-used fitting routine. This is unlikely to be suitable for this data as the 'independent' variable (aAOD) is subject to uncertainties, and AOD typically do not have gaussian error profiles, which are needed for the OLS. Other fitting routines are recommended to be used, like the 'Yorkfit' (York et al., 2004) or a bivariate regression (e.g., Shinozuka et al., 2015). Similarly, some considerations to the "R" parameter should be mentioned – is it the common Pearson linear correlation coefficient or the Spearman's rank correlation as suggested for use in Sayer et al., 2018. It seems uncertain what is used in Matlab's linear model, or how uncertainty is weighted.

7. There seems to be a significant reduction in error statistics when using the Single Oblique angle, than the single nadir view and even the dual views, however this is not mentioned much, and leads the reader to question the validity of the nadir viewing measurements as a result. (see table 1)

8. There seems to be lower discrepancy between syAOD and aAOD in regions of significant biomass burning aerosol (higher AOD Bor, NAW, AOb for example). This raises the question on what type of single scattering albedo is used, and how does the selection of this model impact the AOD retrievals.

9. Throughout the conclusions section there is a significant amount of qualitative wording such as 'agreement is good' This is subjective and not always supported by the comparisons presented in this manuscript. Either give comparison values to what it is expected to be, or refrain from these subjective statements.

10. There is no mention of potential impact of varying single scattering albedo on the AOD retrieval in the conclusion. Is this a solved issue?

*Specific Comments:*

11. Title: 'Extended' seems to be slightly overexaggerating for a year and half in terms of satellite data comparisons. Suggest to remove that word from the title.

12. Line 14: The word 'synergy/synergistic' is used twice in the first sentence.

13. Line 24: The use of double +/- is confusing, is this the error of the error based on AOD, or the potential range of the error?

14. Line 29: Use of "Angström" should be consistent throughout the manuscript, the "ö" is missing on this line.

15. Line 30: AE is not defined .

16. Line 28-35: use of subjective descriptions should be made more quantitative e.g., "good correlation", "agreement is better", "often slightly better". By how much, how often, and compared to what?

17. Abstract: the extent of the evaluation is not introduced. How many days, years, or number of comparison points are used here?

18. Line 108, and throughout the manuscript: there should be a space between the number and the unit '500m'

19. Line 102 and 105, please reference the proper citations for SENTINEL-3 OLCI and SLSTR instead of the websites.

20. Line 102 and subsequent, is it capital case SENTINEL-3, Sentinel-3, Sentinel 3? Please select one and use is consistently throughout the manuscript.

21. Line 113, is there a better reference than this website document for the aerosol retrieval? Seems like this is an important publication for better understanding the material presented in this manuscript. Particularly to support the statement "is of variable quality, with higher uncertainty in retreievals in the oblique backscattering direction." (which has a typo at line 114).
22. Does the shift vectors (section 2.2.1) also have a rotational portion, or is it only translational shifts?
23. Lines 137 and 145 seem to be repeated "at least 50% of valid pixels"
24. Line 147, it is unclear what is meant by 'direction'. Is it viewing direction or viewing angle?
25. Line 151, Does 442.5 spectral band refer to 442.5 nm ?
26. Line 186, What is "Copernicus C3S_Lot2" ?
27. Line 214, why the shift in multiplication symbol from "x" to "*"?
28. Figure 2 is too small.
29. Line 297, How big are the bins in Figure 2?
30. Line 311, GCOS is not defined.
31. Table 1 – decimal point is comma "," instead of point "."
32. Line 342, typo "bind"
33. Line 380, sentence is unclear, is syAOD550 different to S3B $syAOD_{550}$?
34. Line 446, equation 1 does not seem well formatted
35. Line 448, use of * instead of multiplication symbol (×)
36. Line 452, Latitude in [-30 -20] is not well defined, are these degrees south? Is the range inclusive?
37. Line 453-457 , formatting error? dAODrel or is it $dAOD_{rel}$ or dAOD,rel (in figure 8, 9)
38. Line 453, typo? What is "ca"
39. Figure 8, Units on x-axis not identified (Degrees?)
40. Line 547, What is Aerosol_cci+?
41. Line 595, these distribution don't look very Gauss-like, they seem clearly skewed, particularly singleN.
42. Line 617, second apostrophe is not the right side.
43. Line 672-673, portion of this sentence is in red.
44. Figure 22, AOd region is missing a portion of the red dashed curve. (similarly in Figure S10 AsN, and S11 AOb)
45. Line 698, Isn't AERONET reported at 440 -870 nm? What is a personal estimation? AE difference when using a difference in wavelength has been reported in multiple other papers, e.g., LeBlanc et al., 2020, Yoon et al., 2012
46. Figure 24, There seems to be a common clustering of high syAE, at or just above 2.0. Is this a default limit of AE from the retrieval? Or is this a real behavior of the aerosol?
47. Line 735, "good quality" is subjective, but an rms of greater than 0.5, and R often lower than 0.5, with biases often exceeding 1.0 does not seem to be of 'good quality'.
48. Table 7, the decimal notation is a comma "," not a dot "."
49. Figure 28, labels of map regions is too small and of bad quality to read.
50. Line 785, second time AOI is defined.
51. Supplement 1, there is an "Error! Reference source not found." At the 4[th] to last line of the first page.

52. Supplement section 1 and 2, there seems to be no mention of the singleO – oblique angle viewing in the comparison to SURFRAD and Skynet

References:

LeBlanc, S. E., Redemann, J., Flynn, C., Pistone, K., Kacenelenbogen, M., Segal-rosenheimer, M., Shinozuka, Y., Dunagan, S., Dahlgren, R. P., Meyer, K., Podolske, J., Howell, S. G., Freitag, S., Small-griswold, J., Holben, B., Diamond, M., Wood, R., Formenti, P., Piketh, S., Maggs-Kölling, G., Gerber, M. and Namwoonde, A.: Above-cloud aerosol optical depth from airborne observations in the southeast Atlantic, Atmos. Chem. Phys., 20, 1565–1590, doi:10.5194/acp-20-1565-2020, 2020.

Sayer, A. M., Hsu, N. C., Lee, J., Kim, W. V., Burton, S., Fenn, M. A., Ferrare, R. A., Kacenelenbogen, M., LeBlanc, S., Pistone, K., Redemann, J., Segal-Rozenhaimer, M., Shinozuka, Y., and Tsay, S.-C.: Two decades observing smoke above clouds in the south-eastern Atlantic Ocean: Deep Blue algorithm updates and validation with ORACLES field campaign data, Atmos. Meas. Tech., 12, 3595–3627, https://doi.org/10.5194/amt-12-3595-2019, 2019.

Shinozuka, Y., Clarke, A. D., Nenes, A., Jefferson, A., Wood, R., McNaughton, C. S., Ström, J., Tunved, P., Redemann, J., Thornhill, K. L., Moore, R. H., Lathem, T. L., Lin, J. J., and Yoon, Y. J.: The relationship between cloud condensation nuclei (CCN) concentration and light extinction of dried particles: indications of underlying aerosol processes and implications for satellite-based CCN estimates, Atmos. Chem. Phys., 15, 7585–7604, https://doi.org/10.5194/acp-15-7585-2015, 2015.

Yoon, J., Von Hoyningen-Huene, W., Kokhanovsky, A. A., Vountas, M. and Burrows, J. P.: Trend analysis of aerosol optical thickness and ngström exponent derived from the global AERONET spectral observations, Atmos. Meas. Tech., 5(6), 1271–1299, doi:10.5194/amt-5-1271-2012, 2012.

York, D., Evensen, N. M., Martınez, M. L., & De Basabe Delgado, J. Unified equations for the slope, intercept, and standard errors of the best straight line. *American journal of physics*, *72*(3), 367-375., https://doi.org/10.1119/1.1632486, 2004.

---

## Referee Comment (RC3)

Review of revised paper:

**Extended validation and evaluation of the OLCI-SLSTR  synergy aerosol product (SY_2_AOD) on Sentinel-3**          *by L. Sogacheva et al.*

**Highlights**
- now inclusion of fine-mode AOD analysis
- now much better plots on behavior for different AOD regions

**Concerns**
- discussions are too brief (also provide use-recommendations  to potential users?)
- missing comparisons to the standard SLSTR retrieval (to justify the synergy approach)
- no fine-mode AOD results in the abstract and fine-mode AOD comparisons to MODIS
- too extensive  comparisons to AERONET  (move material in the appendix or supplement)
- consider AERONET mid-vis AOD <0.04 (and/or remove mountain AERONET  site data)

**General comments**

The paper investigates  the performance of a combined OLCI and SLSTR  retrievals for AOD, Angstrom (via spectral AOD dependence), AAOD (?), AODf and surface reflection. I assume that the SY_2_AOD retrieval performance mainly mirrors for SLSTR covered regions, the SLSTR retrieval performance with a degraded performance in regions, which only the OLCI sensor covers. Here comparisons and use-statements  are at least needed for the discussion section at the end. The discussion section should also address why anyone (user) would  want to work with SY_2_AOD in comparison to available data from SLSTR (which still have major issues) and especially over available  data from MODIS, VIIRS or MISR.

I could not find a detailed response to my initial review so some of the concerns I voiced in my initial review are still valid. On the other hand, I very much like in the revised version  the new plots that analyze the retrieval performance as function of AOD ranges. These new figures provide much more insights that scatter plots and tables and I suggest to move (the more general performance summaries of) tables (e.g. positive bias but linear fit slope below one seem inconsistent … without the AOD range analysis) and scatter plots - as well as uncertainly analyses into an Appendix or supplement, as the paper is very long and exhausting on the comparisons to AERONET  (e.g. I did not know that spectral surface solar reflection is an official AERONET  product). In that context, I also would focus on AERONET  data with mid-vis AOD > 0.04 (as lower values are likely related to mountain sites, which should not be considered when comparing to regional data (even for regions as small as 3.5x3.5km areas).

Many important regions for aerosol properties have no or only poor AERONET  coverage, so comparisons to global data-sets are essential for a complete pictures. Thus, the effort to compare at the end to a commonly used and likely more mature data-set of MODIS (although potentially  with biases, as MODIS AOD overestimates  over oceans) is well received, but offered comparisons are way too brief and also miss potentially  important AODf comparison (AODf over oceans is offered by the standard MODIS 6.1 product and over land AODf data are available  by MODIS-DB AODdust [AODf ~AOD-AODdust] by Pu, B., Ginoux, P., et al., Atmos. Chem. Phys., 20, 55–81, 2020).

The discussion summary is very brief and disappoints on content, more so since in the data-comparisons, the focus was just on differences and performance with no (or at best little) efforts on interpretations. I strongly suggest to expand the discussion section on major results and their background, so that a reader has a more satisfying element from this comparison paper.

**Specific comments**

27      is there a way to get rid of large outliers (e.g. with a better QA control?)

30      the abstract does not address high AODf bias (for coarse mode dominated references)

119      the aim "to allow for a more robust retrieval" needs to be demonstrated (e.g. vs SLSTR)

119      The aim "to offer data over the entire Sentinel-3 swath" should also be addressed in the discussion (vs SLSTR). I assume similar quality in OCLI only regions over oceans, but significantly reduced quality over OCLI only regions over land.

130      As different products are offered (e.g. all, dual, nadirS, NadirO) are there reasons why particular versions show be used or avoided in particular regions? If the performance all these different SY versions are addressed, there should be some discussion on their use at the end.

177      the Angstrom parameter of the retrieval with AOD at 550 and 865nm over land could be highly inaccurate over vegetation (large/uncertain surf near-IR contributions … any comment?)

295      aAOD as low as 0.02 permitted? (I would use aAOD>0.04 – as a simple way to exlude mountain sites … although a mountain site exclusion to begin with would be better). I suggest to used aAOD >0.04 only.

298      these biases at low AOD are shocking! Why would anyone want to use that product?

305/315  420/424 528/544 654/660      scatter plots and tables (and the explanation) in the Appendix as Figures 3/7/13/19 better tell the entire story.

385      SH Jul-Oct correlation is much better, since (biomass related dry-season) AOD values are higher … so no surprise here.

503      how are uncertainties considered (via weights …?). It is not possible just to remove all data below a specific uncertainty threshold for a higher quality product?

520      what is the value of comparing AOD at longer (865 and 1600nm) wavelengths, when aerosol signals are much weaker (or are completely missed when fine-mode aerosol dominates)?

800      apparent land-sea contrast in SY data (also easily seen in differences to MODIS) need some explanations. I also strongly encourage to extend such comparisons to the AODf for more insights.

870      remove "SKYNET, SURFRAD"

878      the discussion (e.g. "Against MODIS, agreement is good") is way too superficial. MODIS overestimates AOD over oceans (compared to MISR, AATSR and AVHRR-DB … and modeling) so that the relative high SY AOD values over oceans, although they compare to MODIS there) are not really encouraging. A closer inspection will also show that SY AOD – also over oceans – are much more fine-mode dominated than most another satellite retrievals (and modeling), which in part causes the land/ocean contrast of Africa for larger dust outflow AOD.

---

## Author Comment (AC1)

We thank all reviewers for valuable comments and suggestions. After considering them, we see that the quality of the manuscript has improved. Work is planned on further development of the retrieval algorithm based on the validation results reported in the manuscript. Some of the comments/suggestions will be considered during the evaluation of the next version of the Sy AOD product.

**Response to anonymous reviewer #1**

We thank the reviewer for her/his very helpful suggestions

**RC1**: 'Sogacheva et al. (2022) amt-2022-101', Anonymous Referee #2, 03 Jun 2022  reply

**Review for Atmospheric Measurement Techniques**

Title: Extended validation and evaluation of the OLCI-SLSTR Synergy aerosol product (SY_2_AOD) on Sentinel-3

Authors: Larisa Sogacheva, Matthieu Denisselle, Pekka Kolmonen, Timo H. Virtanen, Peter North, Claire Henocq, Silvia Scifoni and Steffen Dransfeld

**General Comments:**

*This manuscript presents a very thorough and detailed validation of the SY_2_AOD and related Angstrom Exponent products by comparison to AERONET and MODIS data sets. This analysis provides the user community with the statistics that are required to intelligently utilize these datasets. What is somewhat lacking in many sections (see some specifics below) are explanations and/or reasons for poor performance in the satellite retrieval AOD products versus AERONET measured AOD in some specific regions. This contrasts with much better performance in other regions yet there is little to no discussion on why some regions are much better than others.*
There are common reasons why the performance of retrieval algorithms is worse at certain conditions (e.g., cloud and snow contamination), in specific regions (e.g., bright surface), and for specific instrument-related reasons (e.g., influence of the viewing geometry, as for S3). Those reasons are mentioned in the text (e.g., lines 154, 205, 865 as in AMTD)
*I think the authors should include much more discussion on the likely algorithmic and/or physical reasons for the discrepancies in the problem regions, much as they did in the last paragraph of the Conclusions section.*
As suggested by all three reviewers, more discussion on the likely algorithmic and/or physical reasons for the discrepancies between Sy_2 and reference products was included.
*Additionally I feel that this paper is too long with too many multi-panel figures for most readers. I suggest that the authors select a significant fraction of the figures (maybe one third) and associated text and move them to an appendix section. This would significantly improve the readability and clarity of the paper.*
One figure and five tables are moved to the Supplement

**Specific Comments:**
*Lines 28-30, Abstract: "The retrieval of Angstrom exponent, related to aerosol size distribution, shows good spatial correlation with expected sources but generally overestimates AE for cases where AERONET Angstrom is low, resulting in overall high bias." I think this somewhat overstates the accuracy and utility of the satellite retrieved AE. The regional AE comparisons in Figure 24 show very poor accuracy for most regions in the satellite AE product. I suggest removing this sentence from the abstract or making a more quantitative statement on the retrieved AE accuracy.*
The statement on the AE is re-formulated.
*Similar comments can be applied to the poor retrieval accuracy of the satellite FMF in Figure 22, except for good agreement at the highest AOD levels.*
Conclusions on the FMF and FMAOD are added to the abstract
*Line 172-173: Please describe somewhere in the text how is AE computed from FMF.*
The section 2.2.3 is now clarified: "During post-processing, further aerosol outputs are derived from the retrieved $AOD_{550}$ and FM AOD. This includes spectral variation of AOD, which is given using pre-computed look-up table

from the retrieved FM AOD and aerosol mixture. The Angstrom exponent is computed based on a pair of spectral AOD values. Here we choose 865nm and 550nm."

*Line 176: Typo, I assume 'duct' is supposed to be 'dust'.*

The typo is corrected

*Line 196-197: Please provide a brief explanation as to why the back scatter at the TOA is more critical in the northern hemispheres versus the southern. Is this just because the percentage of land in the SH is much lower? This is an example of a general lack of physical/algorithm explanations for anomalies and/or comparison results in this manuscript.*

The text for this at line 181 has been rewritten: "In the NH, the SLSTR oblique scan generally samples backscattered radiance, which has a weaker aerosol contribution than the corresponding forward scattering sampled in the SH (e.g., https://www-cdn.eumetsat.int/files/2021-09/SARP_Report_Option_1_final.pdf). This leads to reduced quality in AOD in the Northern Hemisphere (NH) compared with Southern Hemisphere (SH) for the SLSTR products, which has been revealed earlier (https://climate.esa.int/media/documents/Aerosol_cci_PVIR_v1.2_final.pdf). For this reason, SY_2 AOD products from the NH and SH were validated separately."

*Line 250: 'was be' should be 'has been'*

corrected

*Line 265-266: It might be noted that the MAN instruments are calibrated against the same reference instruments as utilized in AERONET. These reference instruments are calibrated by Langley method at Mauna Loa Observatory to an accuracy of 0.002 to 0.005 in the visible and near IR and ~0.009 in the UV.*

The sentence is added, as suggested

*Line 287, Section 6.1: Since AERONET does not measure at 550 nm, please note the spectral interpolation method used. Note that the quadratic or 2nd order fit of AOD versus wavelength is more accurate than the linear or Angstrom fit.*

AE fit was used for interpolation; clarification is added

*Line 295-296: It seems the word 'error' or 'bias' may be missing here. How could 91% of AOD be < 0.04? This AOD level is too low for the majority of the earth.*

The typo (0.04) is corrected to 0.4

*Line 311: Please define the acronym GCOS here.*

GCOS acronym is added

*Line 380: Please provide some reasons or explanation for the smaller retrieval errors in the SH.*

This has been now summarised at line 196 (see earlier comment).

*Line 396-397: An obvious missing region is the Pacific Ocean since oceans dominate the Earth's surface (70%). The Arctic Ocean, Indian Ocean, Southern Ocean, are also very important. Why were these regions not included?*

Validation was performed over Pacific and Indian islands where AERONET stations are located. However, the number of the matchups is critically low over those ocean regions to provide solid conclusion.

*Line 408: It is surprising that the performance is poor for Europe. An explanation of the reason is warranted here.*

For these three comments (408, 409, 414) we feel the correct place to address these discussing is in the conclusion/discussion section, which has now been extended and addresses these points.

*Line 409: The scatter and results for the boreal forest region are very poor. This is surprising since the surface is dark (green forests) and the aerosol type is dominated by fine mode (biomass burning smoke). Please explain/discuss the causes of the poor accuracy retrievals in this region.*

This comment is answered above

*Line 414: An explanation is certainly needed/expected for the large regional differences in the fraction of pixels in EE.*

This comment is answered above

*Line 417-419: The Aus and AOb regions both had very low AOD, none>0.3 so that is a major factor. This should be mentioned in the text otherwise it is somewhat misleading to the reader.*

Details suggested are added to the text

*Lines 444-448, Section 6.1.4: This is an awkward writing style to have a section consist of mainly one line equations and short statements, with no full sentences. I suggest trying to expand it a little to make more readable.*

Section 6.1.4 includes three sub-sections. In the introduction to this section (lines 444-448) we provide only a definition of the relative offset, which is analysed and discussed with respect to different variables (latitude, surface reflectance, ets.) in sub-sections 6.1.4.1-6.1.4.3.

*Line 467-468: In Figure 9 I am missing the separation of NH and SH data that you suggest here. Is there a missing label or legend in this figure?*

There was a typo in the text. The analysis was performed not for the globe, NH and SH, but for dual, singleN and singleO matchups. The sentence was revised

*Line 521-523: The AOD decreases significantly as wavelength increases (except for dust). This may be part of the reason for the offset and rms to decrease as wavelength increases.*

Clarification is added to the text

*There is almost consistently a lack of explanation for the observations/comparisons in this manuscript.*

The main goal of this work (performed in the frame of ESA LAW project) was to evaluate SY_2 AOD product, reveal problems in the retrieval algorithm and notify algorithm developer and potential users about algorithm performance and product quality in different conditions. We also showed that quality is different for different approaches (e.g, dual or single). In case reasons for limited quality were clear (e.g., back scattering contribution, cloud/snow contamination, bright surface), they are mentioned in the text. However, often a throw revision of the retrieval algorithm is needed to find a reason for a limited performance. This work is planned.

*Line 642-643: This is too vague, it does not really say how the AERONET fine mode AOD from SDA was estimated at 550 nm from the 500 nm product. Please provide more detail here.*

A link for the $aAOD_{500}$ to $aAOD_{550}$ conversion is provided

*Line 733-734: A bias in AE of ~1 and rms of 0.5 effectively renders the satellite retrieval of AE as almost useless for most applications. This should be discussed or summarized in the text.*

The AE in table 7 (which is now moved to the supplement) shows consistent positive correlation with AERONET values, albeit with low R values. We see similar patterns in the retrieval of FMF by MODIS as with SLSTR (new Fig. 28), and SLSTR uniquely gives continuous retrieval over land.

*Line 735: By what metric is this syAE considered 'good' quality? I cannot agree with your assessment unless you define 'good' more clearly.*

We move the description of these as 'good', and more simply report the performance.

*Line 740: Validation over ocean: Why are the AE retrievals not compared for over ocean? This would be a useful comparison/validation to include.*

MAN AE (mAE) is provided for 440-870 nm only; Direct comparison between mAE and syAE is not possible

*Line 793-794: Any ideas or explanation about this large difference between MODIS and Sentinel S3A retrievals over Nigeria? This is a striking gradient in large AOD differences, both positive and negative. Which one is more likely to be closer to reality? This is another example of the lack of analysis in giving some explanations in this paper.*

A reason for the large difference is still unclear. We looked at syFMF and modFMF products (new Fig. 27 in the revised version), but modFMF (provided in MOD04_L2 product) is often missing over land. The reason for the luck of explanation is mentioned above (after comment to line 521-523)

*Line 815-816: The way this sentence is written is confusing and does not make too much sense. Please rephrase and clarify.*

The sentence is rephrased

*Line 884-889: This type of analysis and reasons for biases and differences, while good, are mostly lacking in the main text of this paper. It is strange to wait until the Conclusions section to provide this type of analysis.*

We expanded discussion on the reasons for biases and other differences, where reasons for those were clear. To explain some biases, a through revision of the retrieval algorithm is needed.

---

## Author Comment (AC2)

**Response to anonymous reviewer #2**

Thank you very much for your positive review and your helpful comments – they have improved the manuscript greatly.

**Review of "Extended validation and evaluation of the OLCI-SLSTR Synergy aerosol product (SY_2_AOD) on Sentinel-3" by Sogacheva et al.**

*Summary:*

*This paper presents the synergy AOD product from Sentinel-3 and its evaluation against a set of other global AOD products. This is obviously product of a thorough comparison, from the use of validations against AERONET, MAN (and SURFRAD and SKYNET in supplement), and MODIS datasets, and the breadth and level of detail of the manuscript shows it. This is a high-quality manuscript and should be published in AMT, and will likely be used as reference for many other validation of satellite aerosol products. While this manuscript is long, it is obviously needed, and the quality of the work is appreciated.*

*I recommend this paper to be published, but after addressing these issues:*

- *The linear fitting scheme is not well identified, or may not be appropriate for AOD fitting, and by the manuscript's own analysis (section 6.1.5), this matters for quantifying the overall fit. See the general comment #6. This would not be brought up as major concern except for the fact that it is highlighted in the manuscript already.*

Clarification is added to the text that Pearson correlation coefficient was calculated; linear fitting was performed using polynomial. To shorten the manuscript, as requested by the reviewers, we moved results from Sect. 6.1.5 into the Supplement. Link to the Matlab tool for linear fitting considering uncertainties is provided

- *There are numerous errors in formatting throughout the manuscript which detracts from the quality.*

We checked thoroughly AMT requirements for formats and corrected formats accordingly

- *The description of the retrieval methodology is unclear. How does the retrieval of AOD at multiple wavelength and single scattering albedo is achieved through fitting of AOD at only wavelength (550 nm)?*

This is now clarified (line155): We fit both AOD and FMF, which controls the spectral variation of AOD. All wavelengths of SLSTR, and additionally the 442.5nm OLCI channel over land are used in this fitting.

*General Comments:*

1. *Several language issues are found within the abstract, and there is need for more quantitative indication in the abstract instead of the subjective descriptions (see specific comments below)*
   We revised the abstract and provided quantitative indication for the results reported

2. *Throughout the document the date format does not seem to meet the AMT standard of "Date and time: 25 July 2007 (dd month yyyy), 15:17:02 (hh:mm:ss)", particularly evident in the paragraph at line 79-89. See the guidelines: https://www.atmospheric-measurementhttps://www.atmospheric-measurement-techniques.net/submission.html - mathtechniques.net/submission.html#math*
   Date format is corrected in the manuscript according to the AMT standard

3. *How much time is passed between measurements in the oblique and nadir view? And how does that impact the aerosol retrieval, particularly near clouds?*
   To our knowledge, an offset between oblique and nadir view measurements is 1-2 minutes. Cloud screening is performed for both views; cloud edge test is applied

4. *The retrieval dictates the retrieval of AOD and its fine mode at 550 nm, however returns many more parameters, including single scattering albedo, at various wavelengths. This is poorly described, and is*

*both referred to as 'aerosol properties retrieved' and 'intended as diagnostics' (section 2.2.2). Please clarify what these properties are, and how they are retrieved, especially when only fitting to AOD and fine mode AOD at 550 nm.*

This comment is addressed in Sect. 2.2.2

5. *Many references and citations are only links to websites, many of which should be replaced by the appropriate citation, and many are missing the date accessed.*

Most of the links are for technical specifications of the instruments; these links are suggested by ESA as a reference. We checked citations and changed links to the appropriate citations, where possible. However, since S3 is a relatively new mission, not many results are published in the journals. Thus, we refer to the mission documents and results obtained from other projects which are not published yet. If missing, the dates of acceptance are added.

6. *The type of linear regression is not identified, and this matters for AOD comparisons. Reference to a 'linear regression' between the aAOD and syAOD is presented, however it seems to imply the use of the Ordinary-Least-Squares (OLS) commonly-used fitting routine. This is unlikely to be suitable for this data as the 'independent' variable (aAOD) is subject to uncertainties, and AOD typically do not have gaussian error profiles, which are needed for the OLS. Other fitting routines are recommended to be used, like the 'Yorkfit' (York et al., 2004) or a bivariate regression (e.g., Shinozuka et al., 2015). Similarly, some considerations to the "R" parameter should be mentioned – is it the common Pearson linear correlation coefficient or the*

*Spearman's rank correlation as suggested for use in Sayer et al., 2018. It seems uncertain what is used in Matlab's linear model, or how uncertainty is weighted.*

Clarifications for correlation coefficient and linear regression type are added. We agree that linear regression applied to the full range of AOD does describe details and results may be strongly influences by the outliers. Thus, we included in the revised version binned AOD analysis, which shows AOD offset at different AOD ranges.

7. *There seems to be a significant reduction in error statistics when using the Single Oblique angle, than the single nadir view and even the dual views, however this is not mentioned much, and leads the reader to question the validity of the nadir viewing measurements as a result. (see table 1)*

Pixels retrieved with single processor applied to the oblique view are ocean pixels. Retrieved AOD over ocean is, in general, of better quality, because ocean surface reflectance model provides better results that land reflectance approach.

8. *There seems to be lower discrepancy between syAOD and aAOD in regions of significant biomass burning aerosol (higher AOD Bor, NAW, AOb for example). This raises the question on what type of single scattering albedo is used, and how does the selection of this model impact the AOD retrievals.*

This is now clarified (line 157): The SSA is constrained by climatology for the coarse and fine mode extremes separately and as a priori information. The retrieval of FMF results in a SSA by interpolation between these extremes; however, this should be seen as a potential diagnostic for retrieval performance rather than a user product.

9. *Throughout the conclusions section there is a significant amount of qualitative wording such as 'agreement is good' This is subjective and not always supported by the comparisons presented in this manuscript. Either give comparison values to what it is expected to be, or refrain from these subjective statements.*

Statements like 'agreement is good' are accomplished now with values or removed

10. *There is no mention of potential impact of varying single scattering albedo on the AOD retrieval in the conclusion. Is this a solved issue?*

This is included in the conclusion now.

*Specific Comments:*

11. *Title: 'Extended' seems to be slightly overexaggerating for a year and half in terms of satellite data comparisons. Suggest to remove that word from the title.*

We use "extended" not regarding the length of the product, but different validation approaches (including spatial and temporal variations and investigation of the validation results with respect to satellite and

solar geometries) and number of variables which are validated and evaluated (AOD, AODunc, FMAOD, FMF, AE)

12. *Line 14: The word 'synergy/synergistic' is used twice in the first sentence.*

    In the first sentence we explain the origin of the name of the product: the name "synergy" comes from the "synergetic" approach. Thus, the word 'synergy/synergistic' is used twice

13. *Line 24: The use of double +/- is confusing, is this the error of the error based on AOD, or the potential range of the error?*

    The error depends on AOD: for higher AOD, the error envelope is wider

14. *Line 29: Use of "Angström" should be consistent throughout the manuscript, the "ö" is missing on this line.*

    Corrected in the whole manuscript

15. *Line 30: AE is not defined.*

    AE is now defined in the previous line

16. *Line 28-35: use of subjective descriptions should be made more quantitative e.g., "good correlation", "agreement is better", "often slightly better". By how much, how often, and compared to what?*

    Quantitative description (when possible) is added

17. *Abstract: the extent of the evaluation is not introduced. How many days, years, or number of comparison points are used here?*

    Validation period is added to the abstract. Since number of the matchups differs from one exercise to another, depending on the tasks, further datails (e.g., number of validation points) are reported in the main text

18. *Line 108, and throughout the manuscript: there should be a space between the number and the unit '500m'*

    Corrected

19. *Line 102 and 105, please reference the proper citations for SENTINEL-3 OLCI and SLSTR instead of the websites.*

    We used citations recommended by ESA

20. *Line 102 and subsequent, is it capital case SENTINEL-3, Sentinel-3, Sentinel 3? Please select one and use is consistently throughout the manuscript.*

    Checked and corrected in the whole manuscript

21. *Line 113, is there a better reference than this website document for the aerosol retrieval? Seems like this is an important publication for better understanding the material presented in this manuscript. Particularly to support the statement "is of variable quality, with higher uncertainty in retreievals in the oblique backscattering direction." (which has a typo at line 114).*

    The manuscript which describes the retrieval is under preparation. Typo is corrected

22. *Does the shift vectors (section 2.2.1) also have a rotational portion, or is it only translational shifts?*

    Small window (grid) moved around the search window (along shift vectors) in OLCI channel geometry

    (https://sentinels.copernicus.eu/web/sentinel/user-guides/sentinel-3-synergy/definitions/notations)

23. *Lines 137 and 145 seem to be repeated "at least 50% of valid pixels"*

    The text is re-phrased, repetition is removed.

24. *Line 147, it is unclear what is meant by 'direction'. Is it viewing direction or viewing angle?*

    Viewing direction, clarified in the text

25. *Line 151, Does 442.5 spectral band refer to 442.5 nm ?*

    Yes, clarification is provided

26. *Line 186, What is "Copernicus C3S_Lot2" ?*

    We added clarification for the project title, but could not find a proper link to the project description and project documents

27. *Line 214, why the shift in multiplication symbol from "x" to "*"?*

    "*" is replaced with "x" in the whole manuscript

28. *Figure 2 is too small.*

The fonts are corrected

29. *Line 297, How big are the bins in Figure 2?*

Clarification is added to the figure caption

30. *Line 311, GCOS is not defined.*

GCOS is now defined

31. *Table 1 – decimal point is comma ",” instead of point ".”*

Done

32. *Line 342, typo "bind"*

Corrected

33. *Line 380, sentence is unclear, is syAOD550 different to S3B syAOD$_{550}$?*

The sentence is re-phrased

34. *Line 446, equation 1 does not seem well formatted*

Equation 1 is now formatted

35. *Line 448, use of * instead of multiplication symbol ($\times$)*

Corrected. Space between x and aAOD is added, because xaAOD is confusing

36. *Line 452, Latitude in [-30 -20] is not well defined, are these degrees south? Is the range inclusive?*

°S is used now instead of '-'

37. *Line 453-457, formatting error? dAODrel or is it dAOD$_{rel}$ or dAOD,rel (in figure 8, 9)*

In the text, formatting is corrected as it is in figures

38. *Line 453, typo? What is "ca"*

Replaced with ~

39. *Figure 8, Units on x-axis not identified (Degrees?)*

Clarification added to the figure caption

40. *Line 547, What is Aerosol_cci+?*

Link to the project is provided in Sect.2.2

41. *Line 595, these distribution don't look very Gauss-like, they seem clearly skewed, particularly singleN.*

Agree, but it is expected to be Gauss-like

42. *Line 617, second apostrophe is not the right side.*

Corrected

43. *Line 672-673, portion of this sentence is in red.*

Corrected

44. *Figure 22, AOd region is missing a portion of the red dashed curve. (similarly in Figure S10 AsN, and S11 AOb)*

Red-dushed curve is missing in the bins where fine-dominated matchups are missing (blue dots, which are results for fine-dominated matchups are also missing then). However, during the checks, we noticed that the fraction of fine-dominated matchups was calculated from the sum of fine- and coarse- dominated, which is right for AOD binned analysis, but not for FMAOD and FMF analysis, where back-ground matchups may exist in any bin.  This is corrected, fraction of coarse-dominated is added.  Dushed lines for fine- and coarse-dominated fractions are now in blue and green, respectively, as colors for corresponding offsets. The reason for missing a dashed line values at certain bins is the same as it was early – missing fine- or coarse-dominated matchups in the corresponding bin.

45. *Line 698, Isn't AERONET reported at 440 -870 nm? What is a personal estimation? AE difference when using a difference in wavelength has been reported in multiple other papers, e.g., LeBlanc et al., 2020, Yoon et al., 2012*

syAE is reported at 550-870 nm. For evaluation, aAE 500-870 was utilized.

[Figure]

We checked an agreement between $aAE_{440-870}$ and $aAE_{500-870}$ (figure above) and assumed the same agreement between $aAE_{500-870}$ and $aAE_{550-870}$. An offset between $aAE_{440-870}$ and $aAE_{500-870}$ for low (<0.25) AE and high (~2, which is a default value for syAE) AE (which is ~0.2 and ~0.1, respectively) is considerably smaller than an offset between syAE and aAE in those AE size ranges, thus the difference between aAE440-870 and aAE500-870 can be omitted.

46. *Figure 24, There seems to be a common clustering of high syAE, at or just above 2.0. Is this a default limit of AE from the retrieval? Or is this a real behavior of the aerosol?*

    This is a default limit of AE from the retrieval

47. *Line 735, "good quality" is subjective, but an rms of greater than 0.5, and R often lower than 0.5, with biases often exceeding 1.0 does not seem to be of 'good quality'.*

    We made clarification in the text

48. *Table 7, the decimal notation is a comma "," not a dot ".".*

    Corrected

49. *Figure 28, labels of map regions is too small and of bad quality to read.*

    Fonts/labels are corrected

50. *Line 785, second time AOI is defined.*

    Regions for validation with AERONET are defined in Fig.5, Sect.6.3.1. Area of interest for inter-comparison with MODIS is defined in Fig.28 (as in AMTD) and in Table in the Supplement

51. *Supplement 1, there is an "Error! Reference source not found." At the 4th to last line of the first page.*

    The sentence is removed

52. *Supplement section 1 and 2, there seems to be no mention of the singleO – oblique angle viewing in the comparison to SURFRAD and Skynet*

    Low number (or absence) of matchups in group singleO (most pixels in this group are ocean/coastal pixels) did not allow to perform validation with SURFRAD and SKYNET

References:

LeBlanc, S. E., Redemann, J., Flynn, C., Pistone, K., Kacenelenbogen, M., Segal-rosenheimer, M., Shinozuka, Y., Dunagan, S., Dahlgren, R. P., Meyer, K., Podolske, J., Howell, S. G., Freitag, S., Smallgriswold, J., Holben, B., Diamond, M., Wood, R., Formenti, P., Piketh, S., Maggs-Kölling, G., Gerber, M. and Namwoonde, A.: Above-cloud aerosol optical depth from airborne observations in the southeast Atlantic, Atmos. Chem. Phys., 20, 1565–1590, doi:10.5194/acp-20-1565-2020, 2020.

Sayer, A. M., Hsu, N. C., Lee, J., Kim, W. V., Burton, S., Fenn, M. A., Ferrare, R. A., Kacenelenbogen, M., LeBlanc, S., Pistone, K., Redemann, J., Segal-Rozenhaimer, M., Shinozuka, Y., and Tsay, S.-C.: Two decades observing smoke above clouds in the south-eastern Atlantic Ocean: Deep Blue algorithm updates and validation with ORACLES field campaign data, Atmos. Meas. Tech., 12, 3595–3627, https://doi.org/10.5194/amt-12-3595-2019, 2019.

Shinozuka, Y., Clarke, A. D., Nenes, A., Jefferson, A., Wood, R., McNaughton, C. S., Ström, J., Tunved, P., Redemann, J., Thornhill, K. L., Moore, R. H., Lathem, T. L., Lin, J. J., and Yoon, Y. J.: The relationship between

cloud condensation nuclei (CCN) concentration and light extinction of dried particles: indications of underlying aerosol processes and implications for satellite-based CCN estimates, Atmos. Chem. Phys., 15, 7585–7604, https://doi.org/10.5194/acp-15-7585-2015, 2015.

Yoon, J., Von Hoyningen-Huene, W., Kokhanovsky, A. A., Vountas, M. and Burrows, J. P.: Trend analysis of aerosol optical thickness and ngström exponent derived from the global AERONET spectral observations, Atmos. Meas. Tech., 5(6), 1271–1299, doi:10.5194/amt-5-1271-2012, 2012.

York, D., Evensen, N. M., Martınez, M. L., & De Basabe Delgado, J. Unified equations for the slope, intercept, and standard errors of the best straight line. *American journal of physics*, *72*(3), 367-375., https://doi.org/10.1119/1.1632486, 2004.

---

## Author Comment (AC3)

**Response to Stefan Kinne**

We thank Stefan Kinne for giving a positive feedback on the revised (based on his comments) version of the manuscript which was published in AMTD

**Extended validation and evaluation of the OLCI-SLSTR synergy aerosol product (SY_2_AOD) on Sentinel-3** *by L. Sogacheva et al.*

**1 Highlights**

- *now inclusion of fine-mode AOD analysis*
- *now much better plots on behavior for different AOD regions*

**2 Concerns**

- *discussions are too brief (also provide use-recommendations to potential users?)*
  Discussion has been extended.
  Our aim was to perform a critical and detailed evaluation of the Sy_2 product which shows where an improvement of the product is required. Based on this analysis and users needs, they can decide if the product satisfies requirements for their study or not.

- *missing comparisons to the standard SLSTR retrieval (to justify the synergy approach)*
  we reply to this comment below (Specific comments, 119)
- *no fine-mode AOD results in the abstract and fine-mode AOD comparisons to MODIS*
  Results are added
- too extensive comparisons to AERONET
  we aimed to show the performance of the product in different spatial/temporal/geometry conditions and find AERONET is a best choice for that (though we know that AERONET stations are not distributed evenly globally)
- consider AERONET mid-vis AOD <0.04 (and/or remove mountain AERONET site data)
  We answer to this comment below

**3 General comments**

*The paper investigates the performance of a combined OLCI and SLSTR retrievals for AOD, Angstrom (via spectral AOD dependence), AAOD (?), AODf and surface reflection.*

The paper investigates the performance of a combined OLCI and SLSTR retrievals for AOD, AOD uncertainty, Angström (via spectral AOD dependence), AODf and FMF. AAOD and surface reflectance are not among validated/evaluated products.

*I assume that the SY_2_AOD retrieval performance mainly mirrors for SLSTR covered regions, the SLSTR retrieval performance with a degraded performance in regions, which only the OLCI sensor covers.*

AOD in OLCI-only covered areas is not retrieved

*Here comparisons and use-statements are at least needed for the discussion section at the end. The discussion section should also address why anyone (user) would want to work with SY_2_AOD in comparison to available data from SLSTR (which still have major issues) and especially over available data from MODIS, VIIRS or MISR.*

In the current manuscript we aim for evaluating the SY_2 AOD product. Scatter density plots show the presence of outliers (analysis of the outliers, including identification of the location of outliers, is included in LAW validation report); corresponding validation statistics, binned analysis, fraction of matchups in MODIS EE and fraction of matchups which satisfy GCOS requirements show that improvement of the Sy_2 product is needed. Detailed (regional, dust/single retrieval) analysis allows recognition of the conditions in which product quality is better and where an improvement is needed. Based on the validation results, we do not provide recommendations; users can decide if product quality satisfy the requirements for their study (e.g.,for regional analysis) or not.

*I could not find a detailed response to my initial review so some of the concerns I voiced in my initial review are still valid. On the other hand, I very much like in the revised version the new plots that analyze the retrieval performance as function of AOD ranges. These new figures provide much more insights that scatter plots and tables and I suggest to move (the more general performance summaries of) tables (e.g. positive bias but linear fit slope below one seem inconsistent ... without the AOD range analysis) and scatter plots - as well as uncertainly analyses into an Appendix or supplement, as the paper is very long and exhausting on the comparisons to AERONET (e.g. I did not know that spectral surface solar reflection is an official AERONET product).*

Some figures and tables were moved in the supplement. Scatter density plots are left in the main paper, since they show important information, e.g., a distribution of outliers. We also consider that results from the evaluation of provided uncertainties is important for modellers, who exploit AOD uncertainties in models.

*In that context, I also would focus on AERONET data with mid-vis AOD > 0.04 (as lower values are likely related to mountain sites, which should not be considered when comparing to regional data (even for regions as small as 3.5x3.5km areas).*

As suggested, we tested removal of the matchups with aAOD<0.04 from the analysis, but the main results (global, for the NH and SH) have not changed considerably. Thus, we keep old results (for all matchups) in the manuscript.

*Many important regions for aerosol properties have no or only poor AERONET coverage, so comparisons to global data-sets are essential for a complete pictures. Thus, the effort to compare at the end to a commonly used and likely more mature data-set of MODIS (although potentially with biases, as MODIS AOD overestimates over oceans) is well received, but offered comparisons are way too brief and also miss potentially important AODf comparison (AODf over oceans is offered by the standard MODIS 6.1 product and over land AODf data are available by MODIS-DB AODdust [AODf ~AOD-AODdust] by Pu, B., Ginoux, P., et al., Atmos. Chem. Phys., 20, 55–81, 2020).*

We suppose that regions chosen for validation cover most common surface/aerosol conditions globally.

As suggested, we extended FMAOD and FMF evaluation with AERONET.

We also added syFMF and mFMF inter-comparison for test case (26.02.2020).

Validation with ground-based measurements provided valuable information about the product. Extended evaluation with satellite products will be performed when Sy_2 validation results will show a better performance of the retrieval algorithm.

*The discussion summary is very brief and disappoints on content, more so since in the data-comparisons, the focus was just on differences and performance with no (or at best little) efforts on interpretations. I strongly suggest to expand the discussion section on major results and their background, so that a reader has a more satisfying element from this comparison paper.*

The discussion section on major results was expanded. We added interpretations of the results, where reasons for insufficient quality are clear. However, for some results, interpretation is not possible without painstaking testing of the algorithm performance, which is planned to be done based on the validation results. As suggested by another reviewer, subjective conclusions like "good agreement" were accompanied with quantified results.

**4    Specific comments**

**27**            *is there a way to get rid of large outliers (e.g. with a better QA control?)*

AOD quality flags are not provided in the SY_2 product

**30**            *the abstract does not address high AODf bias (for coarse mode dominated references)*

Results are added

**119**            *the aim "to allow for a more robust retrieval" needs to be demonstrated (e.g. vs SLSTR)*

Validation results for the SLSTR v1.12 are not published as a paper yet but available on the CCI web-page.  We provide a link to those results but do not perform an inter-comparison of the SLSTR and SY_2 validation results in the current manuscript for several reasons:

- we aimed at extended validation with high quality ground-based measurements to evaluate the performance of the algorithm in different conditions. We consider that the results presented describe well the status of the product and allow recognition of the "weak" parts in the retrieval algorithm, which helps in the further development of the retrieval algorithm.
- inter-comparison with the SLSTR, if done properly, requires considerable effort (e.g. pixel-to-pixel, retrieved in both products, inter-comparison; repeating SLSTR validation for the same period when Sy_2 product is available, ets.) which was not covered by the tasks in the LAW project
- We agree that an inter-comparison may add additional information. However, the inter-comparison results should be accomplished with a set of figures and discussion, which will extend considerably the current manuscript, which is already long. Detailed inter-comparison with other satellite products may be a subject for another study/manuscript.

Indeed, some validation statistics for the current version of Sy_2 product (retrieval approach follows the main principles and, with some delay, modifications in the SLSTR retrieval algorithm) are slightly worse compared with SLSTR v1.12 product.  However, Sy_2 product is a new product, which is still under development. The validation results reported in the manuscript may also help in further development of the SLSTR AOD product, since the main retrieval approach is simitar for both products.

**119**        *The aim "to offer data over the entire Sentinel-3 swath" should also be addressed in the discussion (vs SLSTR). I assume similar quality in OCLI only regions over oceans, but significantly reduced quality over OCLI only regions over land.*

Over land the retrieval is performed when both SLSTR and OLCI are available

**130**        *As different products are offered (e.g. all, dual, nadirS, NadirO) are there reasons why particular versions show be used or avoided in particular regions? If the performance all these different SY versions are addressed, there should be some discussion on their use at the end.*

One product – Sy_2 AOD – is offered. In this product, AOD is retrieved with two different processors, dual or single, depending on the L1b data availability in nadir and oblique views. Based on flags provided, a user can choose which results (if not all) to use. To help users, we provide validation results for different groups of pixels, combined base on the retrieval approach applied.

*177 the Angstrom parameter of the retrieval with AOD at 550 and 865nm over land could be highly inaccurate over vegetation (large/uncertain surf near-IR contributions ... any comment?)*

We agree that contribution from the vegetation may be a source for AE errors

*295        aAOD as low as 0.02 permitted?  (I would use aAOD>0.04 – as a simple way to exlude mountain sites ... although a mountain site exclusion to begin with would be better). I suggest to used aAOD >0.04 only.*

We answered to this comment in the section "General comments"

*298            these biases at low AOD are shocking!  Why would anyone want to use that product?*

Our aim was to evaluate the first version of the product and show conditions in which a further development of the retrieval algorithm is needed. Users can decide if the quality of the product is enough for their studies (e.g., if they are interested in regional analysis) or not.

*305/315  420/424 528/544 654/660 scatter plots and tables (and the explanation) in the Appendix as Figures 3/7/13/19 better tell the entire story.*

We agree that Figures 3/7/13/19 better tell the story, but not the entire story.  Scatter density plots shows clearly, e.g., the distribution of outliers. This information is hidden in the binned plots (e.g.,  in Fig.3) .

We moved Tables 2-5 into the supplement.

We also added a new figure, where binned offsets (shown also in Fig.2 as magenta dots) for different groups of products (all, dual, singleN, singleO) are combined into one plot (see below). This kind of visualisation shows clearer offsets to AERONET for pixels retrieved with different approaches and the difference in the results for the NH and SH.

[Figure]

*385 SH Jul-Oct correlation is much better, since (biomass related dry-season) AOD values are higher ... so no surprise here.*

Correlation is also good for AOb and Aus, where only low AOD matchups are available

**503** *how are uncertainties considered (via weights ...?). It is not possible just to remove all data below a specific uncertainty threshold for a higher quality product?*

Uncertainties are considered via weights. AOD quality estimate is not provided in the product. In general, uncertainties can be considered as a quality measure, but provided uncertainties for low AOD are often overestimated.

**520** *what is the value of comparing AOD at longer (865 and 1600nm) wavelengths, when aerosol signals are much weaker (or are completely missed when fine-mode aerosol dominates)?*

Often, AE is calculated using $AOD_{865}$. The knowledge on the $AOD_{865}$ quality is important for explanation of the AE quality.

**800** *apparent land-sea contrast in SY data (also easily seen in differences to MODIS) need some explanations. I also strongly encourage to extend such comparisons to the AODf for more insights.*

We included an inter-comparison between syFMF and modFMF (provided in the MOD04_L2 product) for test case described in Sect.8.2 as Fig. 27 in the revised version of the manuscript. Unfortunately, MODIS FMF coverage over land is poor, and thus it can be used for clarification of the difference between syAOD and modAOD.

**870** *remove "SKYNET, SURFRAD"*

We decided to mention SKYNET and SURFRAD here. It is mentioned in the paper (in Introduction) that validation was performed also with SKYNET and SURFRAD and that validation results are provided in the supplement.

**878** *the discussion (e.g. "Against MODIS, agreement is good") is way too superficial. MODIS overestimates AOD over oceans (compared to MISR, AATSR and AVHRR-DB ... and modeling) so that the relative high SY AOD values over oceans, although they compare to MODIS there) are not really encouraging. A closer inspection will also show that SY AOD – also over oceans – are much more fine-mode dominated than most another satellite retrievals (and modeling), which in part causes the land/ocean contrast of Africa for larger dust outflow AOD.*

Validation with MAN shows no bias in SY_ AOD. We add numbers showing the difference between product instead of saying that "agreement is good"